# IMPROVED STOCHASTIC CONTROLLED AVERAGING FOR DISTRIBUTED AND FEDERATED LEARNING

## ABSTRACT

Distributed and federated learning (D/FL) is a powerful machine learning (ML) paradigm in which clients collaborate to train a model under the coordination of a central server. Depending on the nature of clients, data in each client might have the same distribution (called the *homogeneous* setting) or different distributions (the *heterogeneous* setting). The state-of-the-art D/FL algorithm SCAFFOLD addresses the critical issue of data heterogeneity through the use of control variables. However, while theoretical analysis suggests that the convergence rate of SCAFFOLD is independent of data heterogeneity, the practical performance of SCAFFOLD is often inconsistent in homogeneous and heterogeneous settings. Motivated by the disagreement between theory and practice of SCAFFOLD, in this work, we propose a novel D/FL algorithm to bridge this experimental performance gap while preserving similar theoretical guarantees as SCAFFOLD. The proposed algorithm accommodates arbitrary data heterogeneity, partial participation, local updates, and supports unbiased communication compression. Theoretically, we prove that our algorithm is unaffected by data heterogeneity and achieves state-of-the-art convergence rate as SCAFFOLD. Furthermore, numerical experiments indicate that our algorithm achieves consistent (similar) test accuracy in both homogeneous and heterogeneous settings while often converges faster than existing baselines.

## 1 INTRODUCTION

Distributed and federated learning (D/FL) has garnered significant attention due to its effectiveness in the large-scale training of machine learning models (Kairouz et al., 2021). Data used in large-scale training is typically dispersed across a wide variety of clients (or agents). In both settings, a central server is used to orchestrate the local data processing of clients and their collaboration. Under this scheme, the privacy of the clients' data can be maintained as no explicit data is sent from a client to a server (McMahan et al., 2017).

In practice, both paradigms face several key challenges that need to be addressed in algorithmic development. First, not all clients are active at each training step. So, *partial participation* of clients needs to be accounted for in these algorithms. In addition, the communication between clients and the server is often the computational bottleneck, so D/FL algorithms often implement techniques such as local updates and compression to reduce the overall communication costs.

Beyond the above common challenges faced by both distributed training and federated learning, one key difference between these two scenarios is the source of clients (or agents). In distributed training of machine learning (ML) models, high-performance computing resources are abstracted as clients/agents. In this case, data are distributed mainly in order to facilitate parallel computing and to accelerate model training. Therefore, data are distributed in a uniformly random manner, as there is really no difference among those HPC resources. In contrast, clients in federated learning are different in nature, *e.g.*, smartphones and IoT devices (Kairouz et al., 2021). Thus, the data between clients naturally have different distributions, and this *data heterogeneity* has been observed to affect the overall performance of existing algorithms if not accounted for in algorithmic development (Khaled et al., 2020). Therefore, a distributed optimization algorithm that performs consistently well in both homogeneous and heterogeneous settings are in urgent need for distributed training and federated learning.

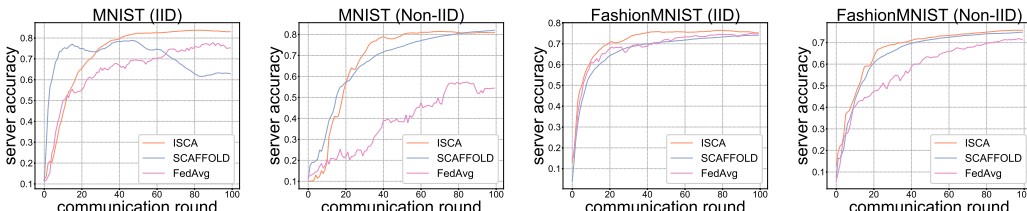

Figure 1: Test accuracy of our algorithm (ISCA), SCAFFOLD, and FedAvg under data homogeneous (IID) and heterogeneous (Non-IID) settings. ISCA converges the fastest and achieves comparable (if not better) test accuracy to FedAvg and SCAFFOLD.

FedAvg (McMahan et al., 2017) (as well as distributed stochastic gradient (DSGD) methods) has emerged as one of the most popular classes of algorithms in distributed training and federated learning. The core idea of FedAvg (and DSGD) is to have clients perform several stochastic gradient descent steps using their local data before a global aggregation period. Despite the simplicity of this scheme, both theoretical and experimental findings show that the performance of FedAvg significantly suffers when the data between clients is heterogeneous (Khaled et al., 2020; Li et al., 2020b). To address this issue, the seminal algorithm SCAFFOLD (Karimireddy et al., 2020) employs local control variables stored on each client alongside a global control variable determined by a central server to correct the "client drift" that otherwise occurs when simply taking stochastic gradient descent updates. Theoretical analysis reveals that the convergence rate of SCAFFOLD is irrespective of the amount of data heterogeneity.

The promising theoretical findings of SCAFFOLD motivate our close study of its practical performance. Unexpectedly, experimental results reveal that the theoretical benefits of SCAFFOLD do not necessarily translate to performance gains in practice. In particular, in the MNIST dataset (see Figure 1), the practical performance of SCAFFOLD is severely inconsistent in homogeneous and heterogeneous settings. This behavior is surprising as theoretically the performance of SCAFFOLD should be completely invariant to data heterogeneity. This observation motivates the following question:

*Can we design a distributed and federated learning (D/FL) algorithm of which both **theoretical** and **practical** performance are unaffected by data heterogeneity?*

Our closer examination of SCAFFOLD allows us to provide a concrete answer to this question and leads to a new algorithm ISCA. First, during one epoch of local updates, SCAFFOLD keeps updating local model parameters while keeping the local control variable *fixed*. This is undesirable as the local control variable is supposed to track the evolvement of the local model. Second, in SCAFFOLD, the local control variable is calculated using an *ancient* local model rather than the newest one. In this work, the aforementioned two observations are leveraged to develop a novel D/FL algorithm that not only possesses the theoretical guarantees of SCAFFOLD but also exhibits consistent practical performance in both homogeneous and heterogeneous settings. We believe this bridges the performance gap between SCAFFOLD and FedAvg we observe in the experiments.

### 1.1 MAIN RESULTS AND CONTRIBUTIONS

The main contributions of this work are summarized below.

- We develop an **I**mproved **S**tochastic **C**ontrolled **A**veraging algorithm (ISCA) for distributed training and federated learning. The proposed algorithm accommodates local updates, partial participation, and arbitrary data heterogeneity. In particular, we highlight two key differences between our algorithm ISCA and SCAFFOLD. First, ISCA updates the local control variable at every local step using the newly computed stochastic gradient. Second, the local control variable clients sent to the server is evaluated at the most recent client model parameter. This is in contrast to SCAFFOLD in which the local control variable is not updated in local steps and is computed using an *ancient* model parameter.

- The theoretical convergence rate of ISCA matches the state-of-the-art result for distributed and federated learning (D/FL) algorithms with arbitrary data heterogeneity; see Table 1 for a brief summary and comparison.

Table 1: Comparison of theoretical convergence rates and practical performance of different FL algorithms. Here, $L$ is the Lipschitz constant of the true gradient, $\sigma^2$ is the variance of the gradient noise, $R$ is related to the initial condition, $N$ is the number of clients, $S$ is the number of active clients, $K$ and $T$ are the number of inner and outer iterations, respectively. The third column lists the additional assumptions needed besides Assumptions 1 and 2.

| Algorithm | Convergence rate | Assumption | Conv. rate vs. heterogeneity (theory) | Practical conv. vs. heterogeneity (practice) |
|---|---|---|---|---|
| FedAvg (McMahan et al., 2017) [a] | $\left(\frac{LR\sigma^2}{NKT}\right)^{1/2} + \left(\frac{LR\zeta}{T}\right)^{2/3} + \frac{LR}{T}$ | Bounded hetero. | ↘ [b] | ↘ |
| VRL-SGD (Liang et al., 2019) | $\left(\frac{LR\sigma^2}{NKT}\right)^{1/2} + \left(\frac{LR\sigma}{\sqrt{K}T}\right)^{2/3} + \frac{LR}{T}$ | – | – | ↘ |
| SCAFFOLD (Karimireddy et al., 2020) | $\left(\frac{LR\sigma^2}{SKT}\right)^{1/2} + \frac{LR}{T}\left(\frac{N}{S}\right)^{2/3}$ | – | – | ↗ [c] |
| ISCA (Theorem 1) | $\left(\frac{LR\sigma^2}{SKT}\right)^{1/2} + \frac{LR}{T}\left(\frac{N}{S}\right)^{2/3}$ | – | – | – |

[a] The constant $\zeta$ is a uniform bound of data heterogeneity, i.e., $\frac{1}{N}\sum_{i=1}^{N}\|\nabla f_i(x) - \nabla f(x)\|^2 \leq \zeta^2$. Other analyses of FedAvg exist, and all rely on certain forms of bounded heterogeneity assumptions.
[b] Worst-case rate is worse when data heterogeneity is more severe.
[c] The practical performance of SCAFFOLD can be much worse in the homogeneous setting.

- To further reduce communication overhead, we incorporate unbiased communication compression into the proposed ISCA algorithm and present ISCAM. We establish theoretical convergence guarantees for ISCAM and show that the convergence rate matches the state-of-the-art result (Huang et al., 2024).

- Numerical experiments are conducted to verify our theoretical findings. Overall, ISCA converges faster than existing baselines. More importantly, the test accuracy achieved by ISCA is consistent in both data homogeneous and heterogeneous settings and matches the theoretical results which states that the convergence rate is independent of the level of heterogeneity.

## 1.2 RELATED WORK

The FedAvg algorithm was first introduced in the seminal work (McMahan et al., 2017) as an algorithm that combines local stochastic gradient updates on clients with a server that performs model averaging. Through extensive empirical studies, McMahan et al. (2017) establishes the effectiveness of FedAvg as an FL algorithm that reduces the number of communication rounds needed to train on decentralized data by orders of magnitude. Inspired by the promising experimental results of FedAvg, extensive efforts have been made to analyze the convergence of FedAvg under various settings. The works (Stich, 2019; Yu et al., 2019; Lin et al., 2020; Wang & Joshi, 2021) analyze FedAvg under the scenario in which the data between agents is homogeneous and all clients participate during the global aggregation period. These analyses are also extended to the more practical setting in which the data between clients are heterogeneous, and only part of the clients participate at each iteration. It has been shown that the presence of data heterogeneity deteriorates the performance of FedAvg because of the "client drift" phenomenon (Li et al., 2020b; Khaled et al., 2020; Karimireddy et al., 2020).

An extensive body of work has been dedicated to addressing the issue of data heterogeneity between clients. FedProx (Li et al., 2020a) adds a proximal term to the objective in order to endow the server with a principled way to account for data heterogeneity. FedNova (Wang et al., 2020) handles data heterogeneity by correctly weighing local models during the global averaging period. FedPD (Zhang et al., 2021) is a meta-algorithm that takes inspiration from primal–dual based algorithms to handle data heterogeneity. FedGATE (Haddadpour et al., 2021) leverages gradient tracking (Xu et al., 2015; Lorenzo & Scutari, 2016; Nedić et al., 2017) to account for the data heterogeneity between clients. The work (Cheng et al., 2024) reveals that FedAvg with momentum can converge without making any assumption that bounds the data heterogeneity even when using a constant stepsize. SCAFFOLD (Karimireddy et al., 2020) is among the most famous FL algorithms and leverages the use of control variables stored on both clients and the server to correct the "client drift" that occurs when clients naively take stochastic gradient updates.

Communication compression has been integrated into FL algorithms to further reduce communication costs. Examples of compressed FL methods include FedPAQ (Reisizadeh et al., 2020), Fed-

Table 2: Comparison of compressed D/FL algorithms. Here, $\omega \in (0, 1]$ is the parameter for compression (see Definition 1), $N \in \mathbb{N}$ is the number of clients, $S \in [N]$ is the number of active clients, and $\epsilon \in \mathbb{R}_{++}$ is the suboptimality measure (*i.e.*, $\mathbb{E}[\|\nabla f(\hat{x})\|^2] \le \epsilon$). Communication complexity is calculated via the convention in Fatkhullin et al. (2024); Huang et al. (2024). Other constants like the Lipschitz constant and the variance $\sigma^2$ of stochastic gradient estimates are omitted for clarity.

| Algorithm | Communication complexity | Partial participation | Data heterogeneity | Standard compressibility |
|---|---|---|---|---|
| FedPAQ (Reisizadeh et al., 2020) | $\frac{N+\omega}{\epsilon S}$ | ✔ | ✗ | ✔ |
| FedCOMGATE (Haddadpour et al., 2021) | $\frac{(1+\omega)N}{\epsilon S}$ | ✗ | ✔ | ✗ |
| VR-MARINA (Gorbunov et al., 2021) | $\frac{\sqrt{N}+\omega}{\epsilon S}$ | ✗ | ✔ | ✔ |
| SCALLION (Huang et al., 2024) | $\frac{(1+\omega)N}{\epsilon S}$ | ✔ | ✔ | ✔ |
| ISCAM (Theorem 1) | $\frac{(1+\omega)N^{2/3}}{\epsilon S^{2/3}}$ | ✔ | ✔ | ✔ |

COMGATE (Haddadpour et al., 2021), SCALLION (Huang et al., 2024). (Here, we focus on the so-called *unbiased* compression; see Definition 1.) Due to the loss of information caused by compression, most of the aforementioned compressed FL methods either lack the robustness to data heterogeneity and partial participation or rely on strict conditions on the compressors; see Table 2 for a brief summary and comparison.

## 2 PROBLEM SETUP

Formally, we consider the following optimization problem in distributed and federated learning:

$$\underset{x \in \mathbb{R}^d}{\text{minimize}} \quad f(x) := \frac{1}{N} \sum_{i=1}^{N} f_i(x), \quad \text{where } f_i(x) := \mathbb{E}_{\xi_i}[F_i(x; \xi_i)],$$

each function $f_i \colon \mathbb{R}^d \to \mathbb{R}$ is smooth but potentially nonconvex, and the symbol $\mathbb{E}_\xi$ denotes the mathematical expectation of the random variable or data $\xi_i$ associated with the probability space $(\mathcal{D}_i, \mathcal{F}_i, \mathbb{P}_i)$. Hence, each $f_i$ is defined as the expected value of some loss function $F_i(\cdot, \xi_i) \colon \mathbb{R}^d \times \mathcal{D}_i \to \mathbb{R}$ over $\xi_i$. The local functions $f_i$ can be different across clients, and such a phenomenon is often called *heterogeneity* in distributed optimization. In federated learning, heterogeneity is often due to the inherent difference in the data distribution $\mathcal{D}_i$ across clients, and this is often referred to as *data heterogeneity*. An undesirable consequence of heterogeneity is that a global stationary point $x^\star$ that satisfies $\nabla f(x^\star) = 0$ may not be a stationary point of any local objective (*i.e.*, $\nabla f_i(x^\star) \neq 0$ for some $i \in [N]$). In contrast, under a *homogeneous* setting, all clients share the same data distribution (*i.e.*, $\mathcal{D}_i = \mathcal{D}$ for all $i \in [N]$) and have the same loss function $f_1 = \cdots = f_N$, so a global stationary point is also stationary for each local objective.

The following standard assumptions are required for our algorithm analyses.

**Assumption 1** (Smoothness). *Each local objective $f_i$ has an L-Lipschitz gradient, i.e., for all $(x, y) \in \mathbf{dom}\, f_i \times \mathbf{dom}\, f_i$ and for all $i \in [N]$, it holds that $\|\nabla f_i(x) - \nabla f_i(y)\| \le L\|x - y\|$.*

**Assumption 2** (Gradient Noise). *There exists $\sigma \in \mathbb{R}_{++}$ such that for all $i \in [N]$ and for all $x \in \mathbf{dom}\, f_i$, it holds that $\mathbb{E}_{\xi_i}[\nabla F_i(x; \xi_i)] = \nabla f_i(x)$ and $\mathbb{E}[\|\nabla F_i(x; \xi_i) - \nabla f_i(x)\|^2] \le \sigma^2$, where $\xi_i \sim \mathcal{D}_i$ are IID random samples for each client $i \in [N]$.*

**Notation.** We use $\mathbb{N} := \{1, 2, \ldots\}$ to denote the set of positive integers and, given $N \in \mathbb{N}$, we denote $[N] := \{1, \ldots, N\}$. We use $\|\cdot\|$ to denote the $\ell_2$ vector norm. The notation $\lesssim$ denotes inequalities that hold up to numeric constants.

## 3 ISCA: IMPROVED STOCHASTIC CONTROLLED AVERAGING FOR FEDERATED LEARNING

In this section, we present the proposed FL algorithm, ISCA, and provide convergence guarantees under the nonconvex, stochastic setup. The development of ISCA is motivated by the practical performance of SCAFFOLD, which is inconsistent in data homogeneous and heterogeneous settings. That being said, we first review the development of SCAFFOLD in §3.1 and study our proposed algorithm ISCA in §3.2. The convergence guarantees of ISCA is then presented in §3.3.

## 3.1 REVIEW OF SCAFFOLD

In order to handle the client drift issue, the SCAFFOLD algorithm Karimireddy et al. (2020) maintains local control variables $\{c_i^t\}_{i=1}^N$ on clients and a global control variable $c^t$ on the server. At each iteration, a set $\mathcal{S}^t$ of active clients is selected to communicate with the server, and then SCAFFOLD conducts $K$ local updates (inner loop) in each active client $i \in \mathcal{S}^t$ by

$$y_i^{t,k+1} \leftarrow y_i^{t,k} + \alpha_{\text{in}}(\nabla F(y_i^{t,k}) - c_i^t + c^t), \quad \text{for } k = 0, \ldots, K-1, \tag{1}$$

where $y_i^{t,k}$ is the local model in client $i$ initialized with the server model $y_i^{t,0} \leftarrow x^t$. This local update can also be viewed as a (stochastic) gradient step *with momentum*. Here, the subscript $i$ represents the client index, and the superscripts $i$ and $k$ denote the iteration counters for the inner loop and outer loop. At the end of local updates, the local control variable is updated as

$$c_i^{t+1} \leftarrow \begin{cases} c_i^t - c^t + \frac{1}{K}(x^t - y_i^{t,K}) & \text{if } i \in \mathcal{S}^t \\ c_i^t & \text{otherwise.} \end{cases} \tag{2}$$

Then, the server updates the global variable (model parameters) and the global control variable as

$$x^{t+1} \leftarrow x^t + \frac{\alpha_{\text{out}}}{S} \sum_{i \in \mathcal{S}^t}(y_i^{t,K} - x^t), \qquad c^{t+1} \leftarrow c^t + \frac{1}{N} \sum_{i \in \mathcal{S}^t}(c_i^{t+1} - c_i^t).$$

Intuitively, the local control variables track the local gradients: $c_i^t \approx \nabla f_i(x^t)$, and similarly, $c^t \approx \nabla f(x^t)$. Consequently, the local updates are nearly synchronized in the presence of data heterogeneity without suffering from client drift.

## 3.2 DEVELOPMENT OF ISCA

Although the theoretical convergence of SCAFFOLD is independent of data heterogeneity Karimireddy et al. (2020), its numerical performance does not seem to be consistent with the theoretical findings. As observed in Figure 1, SCAFFOLD performs much better in the heterogeneous setting than in the homogeneous setting. To fill this gap between theory and practice, we examine the iterations in SCAFFOLD carefully and present our proposed FL algorithm ISCA in Algorithm 1. In particular, ISCA improves upon SCAFFOLD in the following two aspects.

First, we observe that in SCAFFOLD, the updates of local control variables (2) rely on the increment of the local model (*i.e.*, $x^t - y_i^{t,K}$). Also note that the most recent local model $y_i^{t,K}$ is computed using the stochastic gradient evaluated at the previous local iteration (*i.e.*, $\nabla F(y_i^{t,K-1})$); see (1) with $k \leftarrow K-1$. That being said, although Karimireddy et al. (2020) claim that the local control variable in SCAFFOLD tracks the true local gradient (*i.e.*, $c_i^t \approx \nabla f_i(x^t)$), such an approximation is computed using the stochastic gradient evaluated at the previous local model, rather than the *most up-to-date* (stochastic) gradient. Therefore, each active client uploads the *newest* local model and an *ancient* local control variable to the server. This inconsistency might deteriorate the performance of SCAFFOLD. In comparison, the proposed algorithm ISCA performs an additional update of local control variables, using the (stochastic) gradient evaluated at the *newest* local model; see Lines 12 and 13 in Algorithm 1. Modification regarding this aspect is highlighted in red in Algorithm 1, and $v_i^{t,k}$ in ISCA plays a similar role to $c_i^t$ in SCAFFOLD.

Second, SCAFFOLD updates the local model $y_i^{t,k}$ with a *fixed* momentum term $-c_i^t + c^t$. This is undesirable because the local model keeps updating in the inner loop while the local control variable is *fixed*. Although it is often claimed that $c_i^t \approx \nabla f(x^t)$ in SCAFFOLD Karimireddy et al. (2020), the accuracy of such an approximation *should* be improved if the local control variable is updated appropriately in the inner loop. Motivated by this observation, modification regarding this aspect is highlighted in blue in Algorithm 1, and $\{v_i^{t,k}\}_{k=0}^{K+1}$ are the local control variables in ISCA.

## 3.3 CONVERGENCE OF ISCA

This section presents convergence guarantees for Algorithm 1 under the nonconvex, stochastic setup. Remarkably, we only require $L$-smoothness and unbiased, bounded gradient noise.

**Theorem 1.** *Suppose Assumptions 1 and 2 hold, and suppose the stepsizes satisfy* $\alpha = \alpha_{\text{in}}\alpha_{\text{out}}K$, $144\alpha_{\text{in}}^2 K^2 L^2 \leq \alpha$, *and*

$$\alpha \leq \min\left\{\frac{2}{9}, \frac{1}{L}, \left(\frac{KLR}{4ST\sigma^2}\right)^{1/3}, \left(\frac{N^2LR}{33SK\sigma^2}\right)^{1/2}\right\}.$$

---

**Algorithm 1** ISCA: Improved Stochastic Controlled Averaging for Federated Learning

---

**Input:** initial model $x^0 \in \mathbb{R}^d$, global control variable $v^{0,0} \in \mathbb{R}^d$, and local control variables $\{u_i\}_{i=1}^n \subset \mathbb{R}^d$; local (inner loop) learning rate $\alpha_{\mathrm{in}} \in \mathbb{R}_{++}$, global (outer loop) learning rate $\alpha_{\mathrm{out}} \in \mathbb{R}_{++}$, number of local steps $K \in \mathbb{N}_{\geq 1}$; number of sampled clients $S \in [N]$; local dataset $D_i$ on client $i$

1: Initialize $x^0 \in \mathbb{R}^d$, $v^0 \leftarrow \nabla f(x^0)$, and $u_i^{0,0} \leftarrow \nabla f_i(x^0)$ for all $i \in [N]$
2: **for** $t = 0, \ldots, T-1$ **do**
3:      Uniformly sample clients $\mathcal{S}^t \subseteq [N]$ with $|\mathcal{S}^t| = S$
4:      **for** client $i \in \mathcal{S}^t$ in parallel **do**
5:          Receive $x^t$ and $v^t$ from server; initialize $y_i^{t,0} \leftarrow x^t$, $v_i^{t,0} \leftarrow v^t$
6:          **for** $k = 0, \ldots, K-1$ **do**
7:              Compute mini-batch gradient $g_i^{t,k} \leftarrow \nabla F_i(y_i^{t,k}; \xi_i^{t,k})$          $\triangleright\, \xi_i^{t,k} \sim \mathcal{D}_i$
8:              Locally update $y_i^{t,k+1} \leftarrow y_i^{t,k} - \alpha_{\mathrm{in}}(g_i^{t,k} - u_i^{t,k} + v_i^{t,k})$
9:              Locally update $v_i^{t,k+1} \leftarrow v_i^{t,k} + g_i^{t,k} - u_i^{t,k}$
10:            Locally update $u_i^{t,k+1} \leftarrow g_i^{t,k}$
11:          **end for**
12:          $g_i^{t,K} \leftarrow \nabla F(y_i^{t,K}, \xi_i^{t,K})$
13:          $v_i^{t,K+1} \leftarrow v_i^{t,K} + g_i^{t,K} - u_i^{t,K}$
14:          $u_i^{t+1,0} \leftarrow g_i^{t,K}$
15:          Send $y_i^{t,K} - x^t$ and $v_i^{t,K+1}$ to server
16:      **end for**
17:      Update $x^{t+1} \leftarrow x^t + \frac{\alpha_{\mathrm{out}}}{S} \sum_{i \in \mathcal{S}^t}(y_i^{t,K} - x^t)$
18:      Update $v^{t+1} \leftarrow v^t + \frac{1}{N} \sum_{i \in \mathcal{S}^t}(v_i^{t,K+1} - v^t)$
19: **end for**

---

*Then, the $\{x^t\}$ sequence generated by Algorithm 1 satisfies for any $T \in \mathbb{N}$ that*

$$\frac{1}{T} \sum_{t=0}^{T-1} \mathbb{E}[\|\nabla f(x^t)\|^2] \lesssim \sqrt{\frac{LR\sigma^2}{SKT}} + \frac{LR}{T}\left(\frac{N}{S}\right)^{2/3}, \tag{3}$$

*where $R$ is a positive constant related to initial conditions (e.g., $f(x^0) - \inf_x f(x)$).*

**Asymptotic complexities of ISCA.** When the true (full-batch) gradients are used (*i.e.*, $\sigma = 0$), the first term on the right-hand side of (3) vanishes. So, ISCA converges at the classical $\mathcal{O}(1/T)$ rate for first-order optimization methods, and as an instance of FL algorithms, the convergence of ISCA relies on the partial participation rate $N/S$ (see, *e.g.*, Huang et al. (2024)). On the other hand, when the stochastic gradient estimates suffer from a large noise (*i.e.*, $\sigma$ is extremely large), the first term on the right-hand side of (3) dominates. In this case, ISCA enjoys the state-of-the-art $\mathcal{O}(\sigma/\sqrt{T})$ rate, and the performance is mainly hampered by the number of stochastic gradients estimates.

## 4   ISCAM: IMPROVED STOCHASTIC CONTROLLED AVERAGING WITH UNBIASED COMMUNICATION COMPRESSION

Recall that ISCA needs to communicate two quantities (the updates of local models $y_i^{t,K} - x^t$ and the local control variable $v_i^{t,K+1}$) for each client at each iteration. This is slightly undesirable because both FedAvg (McMahan et al., 2017) and SCAFFOLD (Karimireddy et al., 2020; Huang et al., 2024) only require one quantity for uplink communication. To alleviate the additional communication overload in ISCA, we propose to incorporate unbiased communication compression, a technique that has been extensively used in distributed and federated learning methods.

### 4.1   ALGORITHM DESCRIPTION

The development of ISCAM uses the following definition of unbiased communication compressors. This choice of unbiased compressors is standard in the FL literature; see, *e.g.*, Haddadpour

et al. (2021). Examples of compressors that satisfy Definition 1 include random sparsification and dithering; see Appendix A for several examples.

**Definition 1** ($\omega$-unbiased compressor)**.** *For all $x \in \mathbb{R}^d$ and for all client $i \in [N]$, there exist a compressor $\mathcal{C} \colon \mathbb{R}^d \to \mathbb{R}^d$ and a (unified) constant $\omega \in \mathbb{R}_+$ such that*

$$\mathbb{E}_c[\mathcal{C}_i(x)] = x, \qquad \mathbb{E}_c[\|\mathcal{C}_i(x) - x\|^2] \leq \omega\|x\|^2,$$

*where the expectation is taken over the randomness of the compressor $\mathcal{C}_i$, $i \in [N]$.*

**Potential downside of communicating $v_i^{t,K+1}$.** Now, we employ communication compression satisfying Definition 1 into Algorithm 1. One key point in the development of ISCAM is to select the two quantities for uplink communication. Recall that the local control variable $v_i^{t,K+1}$ is designed to better approximate the local gradient evaluated at the global variable; *i.e.*, $v_i^{t,K+1} \approx \nabla f_i(\hat{x})$ when $\hat{x}$ is close to a first-order stationary point. However, due to the (potential) non-IID nature of the data distribution, the local gradient does not converge to zero even when the global gradient converges; *i.e.*, $\nabla f(x^\star) = 0$ but $\nabla f_i(x^\star) \neq 0$. Therefore, if the local control variable $v_i^{t,K+1}$ is compressed, the compression error introduced at each communication round satisfies

$$\mathbb{E}_c[\|\mathcal{C}_i(\nabla f_i(x^t)) - \nabla f_i(x^t)\|^2] \leq \omega\|\nabla f_i(x^t)\|^2 \to \omega\|\nabla f_i(x^\star)\|^2 \neq 0,$$

which implies that the compression error might be very large under severe data heterogeneity.

**Benefits of compressing $v_i^{t,K+1} - v^t$.** One remedy for this above issue is to communicate the increment of the local control variable, *i.e.*, $v_i^{t,K+1} - v^t$. Adding together the $v_i^{t,k+1}$-update (Line 9 in Algorithm 1) gives $v_i^{t,K+1} - v^t = g_i^{t,K} - u_i^{t,0} = u_i^{t+1,0} - u_i^{t,0} =: \Delta_i^t$. (Recall that $v_i^{t,0} = v^t$ by initialization.) Since $u_i^{t,0}$ is the *cached* local gradient, ideally $u_i^{t,0} \to \nabla f_i(x^\star)$ if $x^t \to x^\star$. Therefore, the increment control variable $\Delta_i^t$ vanishes eventually, and compressing $\Delta_i^t$ results in a vanishing compression error $\mathbb{E}_c[\|\mathcal{C}_i(\Delta_i^t) - \Delta_i^t\|^2] \leq \omega\|\Delta_i^t\|^2 \to 0$, regardless of data heterogeneity.

Following the above discussion, we are ready to incorporate unbiased compression into ISCA (Algorithm 1) and present the resulting algorithm (ISCAM) in Algorithm 2. The parameters $(\beta_1, \beta_2) \in [0,1]^2$ are introduced to stabilize the updates of the increment variables and can be viewed as learning rates. When $\beta_1 = \beta_2 = 1$ and $\{\mathcal{C}_i\}_{i=1}^N$ are the identity mappings (*i.e.*, no compression), ISCAM reduces to ISCA.

## 4.2 CONVERGENCE ANALYSIS OF ISCAM

We now present the convergence results for Algorithm 2 under the nonconvex, stochastic setup and with an unbiased compressor that satisfies Definition 1.

**Theorem 2.** *Suppose Assumptions 1 and 2 hold, and suppose the stepsizes satisfy $\alpha = \alpha_{\text{in}}\alpha_{\text{out}}K$, $144\alpha_{\text{in}}^2 K^2 L^2 \leq \alpha$, $\beta_1 = \frac{1-\alpha}{4}$, $\beta_2 = \frac{\alpha SL}{27N}$, and*

$$\alpha \leq \min\left\{\frac{2}{9(1+\omega)}, \frac{1}{L}, \left(\frac{KLR}{4(1+\omega)ST\sigma^2}\right)^{1/3}, \left(\frac{N^2LR}{50(1+\omega)SK\sigma^2}\right)^{1/2}\right\}.$$

*Then, the $\{x^t\}$ sequence generated by Algorithm 2 satisfies for any $T \in \mathbb{N}$ that*

$$\frac{1}{T}\sum_{t=0}^{T-1}\mathbb{E}[\|\nabla f(x^t)\|^2] \lesssim \sqrt{\frac{(1+\omega)LR\sigma^2}{SKT}} + \frac{(1+\omega)LR}{T}\left(\frac{N}{S}\right)^{1/3} + \left(\frac{(1+\omega)N^2L^2R^2\sigma^2}{S^3KT^2}\right)^{1/3}$$

*where $R$ is a positive constant related to initial conditions (e.g., $f(x^0) - \inf_x f(x)$).*

A detailed version and the proof are presented in Appendix D.

**Computation complexity of ISCAM.** The computation complexity of ISCAM is similar to that of ISCA. When no gradient noise exists (*i.e.*, $\sigma \to 0$), the convergence of ISCAM has the $\mathcal{O}(1/T)$ sublinear rate and is also restricted by the partial participation ratio $N/S$. On the other hand, when $\sigma$ is extremely large and the terms involving $\sigma$ are dominating, the $\sigma/\sqrt{T}$-dependent term dominates others. Again, in this case, the performance is mainly hampered by the number of gradient evaluations, which is the same as the convergence result for ISCA.

---

**Algorithm 2** ISCAM: Improved Stochastic Controlled Averaging with Unbiased Compression

---

**Input:** initial model $x^0 \in \mathbb{R}^d$, global control variable $v^{0,0} \in \mathbb{R}^d$, and local control variables $\{u_i\}_{i=1}^n \subset \mathbb{R}^d$; local (inner loop) learning rate $\alpha_{\text{in}} \in \mathbb{R}_{++}$, global (outer loop) learning rate $\alpha_{\text{out}} \in \mathbb{R}_{++}$, number of local steps $K \in \mathbb{N}_{\geq 1}$; number of sampled clients $S \in \mathbb{N}_{\geq 1}$; local dataset $\mathcal{D}_i$ on client $i$

1: **for** $t = 0, \dots, T-1$ **do**
2:      Uniformly sample clients $\mathcal{S}^t \subseteq [N]$ with $|\mathcal{S}^t| = S$
3:      **for** client $i \in \mathcal{S}^t$ in parallel **do**
4:          Receive $x^t$ and $v^t$ from server; initialize $y_i^{t,0} \leftarrow x^t, v_i^{t,0} \leftarrow v^t$
5:          **for** $k = 0, \dots, K-1$ **do**
6:              Compute mini-batch gradient $g_i^{t,k} \leftarrow \nabla F_i(y_i^{t,k}; \xi_i^{t,k})$        $\triangleright \xi_i^{t,k} \sim \mathcal{D}_i$
7:              Locally update $y_i^{t,k+1} \leftarrow y_i^{t,k} - \alpha_{\text{in}}(g_i^{t,k} - u_i^{t,k} + v_i^{t,k})$
8:              Locally update $v_i^{t,k+1} \leftarrow v_i^{t,k} + g_i^{t,k} - u_i^{t,k}$
9:              Locally update $u_i^{t,k+1} \leftarrow g_i^{t,k}$
10:          **end for**
11:          $g_i^{t,K} \leftarrow \nabla F(y_i^{t,K}, \xi_i^{t,K})$
12:          $v_i^{t,K+1} \leftarrow v_i^{t,K} + g_i^{t,K} - u_i^{t,K}$
13:          Compute $\delta_i^t \leftarrow \frac{\beta_1(y_i^{t,K} - x^t)}{\alpha_{\text{in}} K}$ and $\Delta_i^t \leftarrow \beta_2(v_i^{t,K+1} - v^t)$
14:          $u_i^{t+1,0} \leftarrow u_i^{t,0} + \Delta_i^t$
15:          Compress and send $\tilde{\delta}_i^t \leftarrow \mathcal{C}_i(\delta_i^t)$ and $\tilde{\Delta}_i^t \leftarrow \mathcal{C}_i(\Delta_i^t)$ to the server
16:      **end for**
17:      Update $x^{t+1} \leftarrow x^t + \frac{\alpha_{\text{out}} \alpha_{\text{in}} K}{S} \sum_{i \in \mathcal{S}^t} \tilde{\delta}_i^t$
18:      Update $v^{t+1} \leftarrow v^t + \frac{1}{N} \sum_{i \in \mathcal{S}^t} \tilde{\Delta}_i^t$
19: **end for**

---

**Asymptotic communication complexity of ISCAM.** Following the convention in Fatkhullin et al. (2024); Huang et al. (2024), the asymptotic communication complexity is defined as the total number of communication rounds required to obtain $\mathbb{E}[\|\nabla f(\hat{x})\|^2]$ in the regime $\sigma \to 0$. Then, Theorem 2 shows $\frac{1+\omega}{\epsilon}(\frac{N}{S})^{1/3}$ asymptotic communication complexity. (It is derived from $\frac{(1+\omega)LR}{T}(\frac{N}{S})^{1/3} \asymp \epsilon$.) So, ISCAM improves the influence of the client participation ratio on the asymptotic communication complexity for compressed FL methods with non-IID clients. Regarding the impact of stationarity measure $\epsilon$ and the compression parameter $\omega$, ISCAM matches the state-of-the-art asymptotic communication complexity and does not require a uniform bound on the compression errors (as did in many compressed FL methods with non-IID clients; see, *e.g.*, Haddadpour et al. (2021)).

Based on the above discussion, we demonstrate that ISCAM theoretically improves existing FL methods with unbiased compression.

## 5 EXPERIMENTS

This section presents numerical experiments to demonstrate the efficacy of the proposed methods. Recall that this work is motivated by the performance gap of SCAFFOLD between theory and practice. So in the experiments, we aim to demonstrate that the proposed algorithms have consistent (*i.e.*, similar) practical performance in both data homogeneous and heterogeneous settings, which aligns with our theoretical findings (Theorems 1 and 2).

### 5.1 EXPERIMENTAL SETTINGS

**Datasets and baselines.** We use two standard FL benchmark datasets: MNIST (Deng, 2012) and Fashion MNIST (Xiao et al., 2017) in the numerical experiments. Both MNIST and Fashion MNIST have 60,000 training images and 10,000 test images, each of which is categorized into one of ten classes. A (nonconvex) fully connected neural network is used as the model in the experiments, and following the convention in Karimireddy et al. (2020), it consists of two hidden layers (256 and 128 neurons for each layer). ISCA is compared with SCAFFOLD (Karimireddy et al., 2020) and FedAvg (McMahan et al., 2017), two state-of-the-art FL algorithms in the data homogeneous and

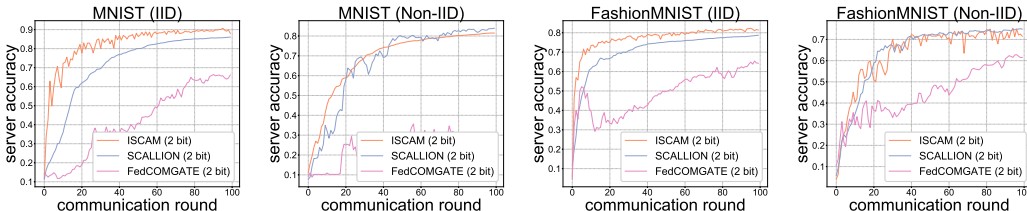

Figure 2: Test accuracy of ISCAM, SCALLION, and FedCOMGATE under data homogeneous (IID) and heterogeneous (non-IID) settings, with 2-bit compression, and tested on MNIST (left half) and Fashion MNIST (right half).

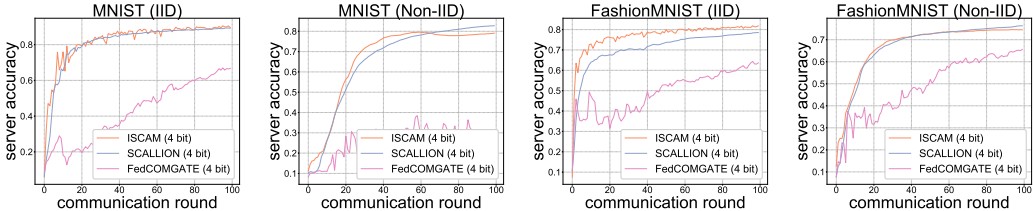

Figure 3: Test accuracy of ISCAM, SCALLION, and FedCOMGATE under data homogeneous (IID) and heterogeneous (non-IID) settings, with 4-bit compression, and tested on MNIST (left half) and Fashion MNIST (right half).

heterogeneous settings, respectively. For FL algorithms with compression, we compare the proposed ISCAM with SCALLION (Huang et al., 2024) and FedCOMPACT (Haddadpour et al., 2021).

**Experimental settings.** In the experiments, the training data are distributed across $N = 100$ clients. In the homogeneous (IID) setting, all the training data are distributed in all the clients in a uniformly random manner, while in the heterogeneous (non-IID) setting, the dataset is split into 200 shards, each containing samples from only one class. (Each client is assigned with two shards randomly.) At each training epoch, the server selects $S = 10$ clients in a random manner, and the sampled clients take $K = 5$ local steps to update their own model parameters. In the experiments, we tune the global learning rate ($\alpha_{\text{out}}$) and the local learning rate ($\alpha_{\text{in}}$) over the grid $\{0.01, 0.05, 0.1, 0.5, 1, 3, 5, 10\}^2$. For compressed FL algorithms, we use the unbiased random dithering compression (Alistarh et al., 2017). For ISCAM, the additional learning rates are set to $\beta_1 = \beta_2 = 0.1$. In this section, we use test accuracy as the performance metric, following standard FL literature. Additional experimental results are in Appendix E.

## 5.2 NUMERICAL RESULTS

**Numerical results for ISCA.** In Figure 1, we present the test accuracy of the proposed ISCA, SCAFFOLD (Karimireddy et al., 2020), and FedAvg (McMahan et al., 2017). We observe that on both datasets, ISCA achieves similar test accuracy in the data homogeneous and heterogeneous settings, consistent with our theoretical findings (Theorem 1). FedAvg performs much better in the homogeneous setting than the heterogeneous setting, which has been extensively studied (McMahan et al., 2017). As expected, SCAFFOLD performs quite well in the heterogeneous setting. The unexpected result is that the performance of SCAFFOLD sometimes deteriorates significantly in the homogeneous settings. This observation contradicts all the existing theoretical results (Karimireddy et al., 2020; Huang et al., 2024) which claim that the convergence of SCAFFOLD is irrespective of the amount of data heterogeneity. Moreover, we see that in most cases, ISCA converges faster than SCAFFOLD and FedAvg, further demonstrating the superiority of the proposed algorithm.

**Numerical results for ISCAM.** In Figures 2 and 3, we plot the same set of experimental results and compare the proposed ISCAM with SCALLION (Huang et al., 2024) and FedCOMGATE (Haddadpour et al., 2021), applying unbiased random dithering (Alistarh et al., 2017) with 2-bit and 4-bit per entry, respectively. Similarly, we see that ISCAM performs consistently well (in terms of test accuracy) in both homogeneous and heterogeneous settings. In terms of convergence rate, ISCAM has comparable, sometimes better, practical convergence speed compared with SCALLION.

# 6 CONCLUSION

This paper presents ISCA, a new distributed and federated learning (D/FL) algorithm that admits local updates, partial participation, and arbitrary data heterogeneity. Theoretically, under minimal assumptions, ISCA matches the state-of-the-art convergence of D/FL methods. Empirically, ISCA converges faster than other baselines and achieves consistent test accuracy in the data homogeneous and heterogeneous settings.

Moreover, to further reduce the communication overload, we incorporate unbiased communication compression into ISCA and propose a new compressed FL algorithm called ISCAM. Theoretical convergence of ISCAM is established under standard compressibilities and matches (or improves) the state-of-the-art result for compressed FL methods. Numerical experiments also support our theoretical findings on ISCAM.

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

## A    EXAMPLES OF UNBIASED COMPRESSORS

In this section, we present two examples of unbiased compressors.

**Example 1** (Random sparsification). *For any $s \in [d]$, the random-$s$ sparsification is defined as $\mathcal{C} \colon x \mapsto \frac{d}{s}(\xi \odot x)$, where $\odot$ is the Hadamard (entry-wise) product and $\xi \in \{0,1\}^d$ is a uniformly random binary vector with $s$ nonzero entries. This random-$s$ sparsification is an unbiased compressor with parameter $\omega = \frac{d}{s} - 1$.*

**Example 2** (Random dithering (Alistarh et al., 2017)). *For any $b \in \mathbb{N}_+$, the random dithering with $b$-bits per entry is defined as $\mathcal{C} \colon x \mapsto \|x\| \times \mathbf{sign}(x) \odot \zeta(x)$, where $\{\zeta_k\}_{k=1}^d$ are independent random variables such that*

$$
\zeta_k(x) := \begin{cases} \left\lfloor \frac{2^b |x_k|}{\|x\|} \right\rfloor & \text{with probability } \left\lceil \frac{2^b |x_k|}{\|x\|} \right\rceil - \frac{2^b |x_k|}{\|x\|} \\[2ex] \left\lceil \frac{2^b |x_k|}{\|x\|} \right\rceil & \text{otherwise,} \end{cases}
$$

*where $\lfloor \cdot \rfloor$ and $\lceil \cdot \rceil$ are the floor and ceiling functions, respectively. This random dithering with $b$-bits per entry is an unbiased compressor with parameter $\omega = \min\left\{\frac{d}{4^b}, \frac{\sqrt{d}}{2^b}\right\}$.*

## B    PRELIMINARIES FOR CONVERGENCE ANALYSIS

### B.1    NOTATION

We abbreviate $\sum_{i=1}^N$, $\sum_{k=0}^{K-1}$, $\sum_{i=1}^N \sum_{k=0}^{K-1}$ as $\sum_i$, $\sum_k$, $\sum_{i,k}$, respectively, when no confusion occurs. We also define the quantities

$$
g_i^t := \frac{1}{K} \sum_{k=0}^{K-1} g_i^{t,k}, \qquad g^t := \frac{1}{N} \sum_{i=1}^N g_i^t, \qquad g^{t,K} := \frac{1}{N} \sum_i g_i^{t,K}, \qquad v_i^t := \frac{1}{K} \sum_{k=0}^{K-1} v_i^{t,k},
$$

$$
X^t := \mathbb{E}[\|x^t - x^{t-1}\|^2], \qquad Y^t := \frac{1}{NK} \sum_{i=1}^N \sum_{k=1}^{K-1} \mathbb{E}[\|y_i^{t,k} - x^t\|^2], \tag{4}
$$

$$
U^t := \frac{1}{N} \sum_{i=1}^N \mathbb{E}[\|u_i^{t,0} - \nabla f_i(x^{t-1})\|^2], \qquad V^t := \mathbb{E}[\|v^t - \nabla f(x^{t-1})\|^2].
$$

In addition, we denote $\alpha := \alpha_{\text{in}} \alpha_{\text{out}} K$ and

$$
d^{k+1} := \tfrac{1}{\alpha}(x^t - x^{t+1}) \equiv \tfrac{1}{S} \sum_{i \in \mathcal{S}^t} \left(v_i^t + \tfrac{1}{K}(g_i^{t,K-1} - u_i^{t,0})\right),
$$

which follows from the $x$-update (Line 17 in Algorithm 1 or Line 17 in Algorithm 2).

### B.2    PRELIMINARY RESULTS

In this section, we present some preliminary inequalities that will be used in our analysis. Most of them are established results in the literature and are irrelevant to any specific optimization algorithms.

**Lemma 1.** *For all $\theta \in [0, \frac{1}{2}]$ and for all $(a, b) \in \mathbb{R}^n \times \mathbb{R}^n$, it holds that*

$$
(1 - 2\theta)\|a - b\|^2 \le (1 - \theta)\|a\|^2 + \frac{1}{\theta}\|b\|^2.
$$

*Proof.* When $\theta \in [0, \frac{1}{2}]$, we have $1 - 2\theta \ge 0$ and thus

$$
(1 - 2\theta)\|a - b\|^2 = (1 - 2\theta)(\|a\|^2 + \|b\|^2 - 2a^T b)
$$

$$
= (1 - 2\theta)(\|a\|^2 + \|b\|^2) - 2(1 - 2\theta)a^T b
$$

$$
\le (1 - 2\theta)(\|a\|^2 + \|b\|^2) + (\theta\|a\|^2 + \tfrac{(1-2\theta)^2}{\theta}\|b\|^2)
$$

$$= (1-\theta)\|a\|^2 + \tfrac{(1-2\theta)(1-\theta)}{\theta}\|b\|^2$$

$$\leq (1-\theta)\|a\|^2 + \tfrac{1}{\theta}\|b\|^2.$$

The first inequality uses the Young's inequality $2a^\mathsf{T}b \leq \rho\|a\|^2 + \tfrac{1}{\rho}\|b\|^2$ with $\rho \leftarrow \tfrac{\theta}{1-2\theta}$. The last inequality follows from the fact that $\theta \in [0, 1]$ and thus $(1 - 2\theta)(1 - \theta) = 1 + \theta(2\theta - 3) \leq 1$. $\qquad\square$

**Lemma 2** (Cheng et al. (2024, Lemma 7)). *Given any $\{a_i\}_{i=1}^N \subset \mathbb{R}^d$ and $b \in \mathbb{R}^d$, denote $a := \tfrac{1}{N}\sum_{i\in[N]} a_i$. Suppose $\mathcal{S} \subset [N]$ is uniformly sampled from $[N]$ without replacement with $|\mathcal{S}| = S$. Then, it holds that*

$$\mathbb{E}_\mathcal{S}\left[\left\|\frac{1}{S}\sum_{i\in\mathcal{S}} a_i\right\|^2\right] \leq \|a\|^2 + \frac{1}{SN}\sum_i \|a_i - a\|^2 \leq \|a\|^2 + \frac{1}{SN}\sum_i \|a_i - b\|^2,$$

*where the expectation is taken over all the possible sampling of $\mathcal{S}$.*

**Lemma 3** (Karimireddy et al. (2020, Lemma 4)). *Let $\{X_1, \cdots, X_\tau\} \subset \mathbb{R}^d$ be random variables that are potentially dependent. If their means and variances satisfy $\mathbb{E}[X_i] = \mu_i$ and $\mathbb{E}[\|X_i - \mu_i\|^2] \leq \sigma^2$, then it holds that*

$$\mathbb{E}\left[\left\|\sum_{i=1}^\tau X_i\right\|^2\right] \leq \left\|\sum_{i=1}^\tau \mu_i\right\|^2 + \tau^2\sigma^2.$$

*If instead their means and variances satisfy $\mathbb{E}[X_i|X_{i-1}, \ldots, X_1] = \mu_i$ and $\mathbb{E}[\|X_i - \mu_i\|^2] \leq \sigma^2$, then it holds that*

$$\mathbb{E}\left[\left\|\sum_{i=1}^\tau X_i\right\|^2\right] \leq 2\mathbb{E}\left[\left\|\sum_{i=1}^\tau \mu_i\right\|^2\right] + 2\tau\sigma^2.$$

**Lemma 4** (Huang et al. (2024, Lemma 3)). *Under Assumptions 1 and 2, for all $\theta \in [0, 1]$, $w \in \mathbb{R}^d$ and $\{w_i\}_{i=1}^N \subset \mathbb{R}^d$, the sequence $\{x^t\}$ generated by Algorithm 1 (or Algorithm 2) and the sequence $\{g^t\}$ defined in (4) satisfy*

$$\mathbb{E}[\|(1-\theta)w + \theta(g^t - \nabla f(x^t))\|^2]$$

$$\leq \min\left\{2(1-\theta)\mathbb{E}[\|w\|^2] + 3\theta^2L^2Y^t + \tfrac{2\theta^2\sigma^2}{NK}, (1-\theta)\mathbb{E}[\|w\|^2] + 2\theta L^2Y^t + \tfrac{2\theta\sigma^2}{NK}\right\}, \quad (5)$$

*and*

$$\tfrac{1}{N}\sum_i \mathbb{E}[\|(1-\theta)w_i + \theta(g_i^t - \nabla f_i(x^t))\|^2]$$

$$\leq \min\left\{\tfrac{2(1-\theta)}{N}\sum_i \mathbb{E}[\|w_i\|^2] + 3\theta^2L^2Y^t + \tfrac{2\theta^2\sigma^2}{K}, \tfrac{1-\theta}{N}\sum_i \mathbb{E}[\|w_i\|^2] + 2\theta L^2Y^t + \tfrac{2\theta\sigma^2}{K}\right\}, \quad (6)$$

*where the quantity $Y^t$ is defined in (4).*

## C  CONVERGENCE ANALYSIS FOR ALGORITHM 1

We start with two fundamental pillars for the convergence analysis. In particular, Lemma 5 establishes the key connection between the global tracking variable $v^t$ and local tracking variable $u_i^{t,0}$, and Lemma 6 provides bounds on the stochastic gradients $g_i^t$, $g_i^{t,K}$ and $g^{t,K}$. The important descent inequality is then provided in Lemma 7.

**Lemma 5.** *For all $t \in \mathbb{N}$, the sequence $\{v^t\}$ generated by Algorithm 1 satisfies*

$$v^t = \frac{1}{N}\sum_{i=1}^N u_i^{t,0}. \tag{7}$$

*In addition, for all $t \in \mathbb{N}$ and $i \in [N]$, the sequence $\{v_i^t\}$ defined in (4) satisfies*

$$v_i^t = v^t + \frac{1}{K}\sum_{k=0}^{K-2}\left(g_i^{t,k} - (K-1)u_i^{t,0}\right). \tag{8}$$

*Proof.* We prove the first statement via mathematical induction.

1. In the base case where $t = 0$, by initialization it holds that $v^0 = \frac{1}{N} \sum_i u_i^{0,0}$.

2. The induction hypothesis assumes that $v^t = \frac{1}{N} \sum_i u_i^{t,0}$ for any $t \in \mathbb{N}$.

3. The induction step: we prove that $v^{t+1} = \frac{1}{N} \sum_i u_i^{t+1,0}$. It follows from the $v_i^{t,k+1}$-update (Line 9 in Algorithm 1) that

$$v_i^{t,K+1} = v_i^{t,0} + \sum_k (g_i^{t,k} - u_i^{t,k}) = v_i^{t,0} + g_i^{t,K} - u_i^{t,0} = v^t + u_i^{t+1,0} - u_i^{t,0}.$$

Then, it follows from the $v$-update (Line 18 in Algorithm 1) that

$$
\begin{aligned}
v^{t+1} &= v^t + \frac{1}{N} \sum_{i \in \mathcal{S}^t} (v_i^{t,K+1} - v^t) \\
&= \frac{1}{N} \sum_i u_i^{t,0} + \frac{1}{N} \sum_{i \in \mathcal{S}^t} (u_i^{t+1,0} - u_i^{t,0}) \\
&= \frac{1}{N} \sum_{i \in \mathcal{S}^t} u_i^{t+1,0} + \frac{1}{N} \sum_{i \notin \mathcal{S}^t} u_i^{t,0} \\
&= \frac{1}{N} \sum_{i \in \mathcal{S}^t} u_i^{t+1,0} + \frac{1}{N} \sum_{i \notin \mathcal{S}^t} u_i^{t+1,0} \\
&= \frac{1}{N} \sum_i u_i^{t+1,0}.
\end{aligned}
$$

Therefore, the desired result (7) follows from mathematical induction.

For the second argument, it follows from Line 9 that

$$
\begin{aligned}
v_i^{t,0} &= v_i^{t,0} \\
v_i^{t,1} &= v_i^{t,0} + g_i^{t,0} - u_i^{t,0} \\
v_i^{t,2} &= v_i^{t,0} + (g_i^{t,0} - u_i^{t,0}) + (g_i^{t,1} - g_i^{t,0}) \\
&\vdots \\
v_i^{t,K-1} &= v_i^{t,0} + (g_i^{t,0} - u_i^{t,0}) + (g_i^{t,1} - g_i^{t,0}) + \cdots + (g_i^{t,K-2} - g_i^{t,K-3}) \\
\sum_{k=0}^{K-1} v_i^{t,k} &= K v_i^{t,0} + \left( -(K-1) u_i^{t,0} + g_i^{t,0} + g_i^{t,1} + \cdots + g_i^{t,K-2} \right) \\
&= K v^t + \sum_{k=0}^{K-2} \left( g_i^{t,k} - (K-1) u_i^{t,0} \right),
\end{aligned}
$$

and then the desired result (8) follows from the definition $v_i^t := \frac{1}{K} \sum_k v_i^{t,k}$. $\qquad \square$

**Lemma 6.** *Under Assumptions 1 and 2, it holds for all $t \in \mathbb{N}$ that*

$$\frac{1}{N} \sum_i \mathbb{E}[\|g_i^t - u_i^{t,0}\|^2] \leq 4L^2 X^t + 4L^2 Y^t + 4U^t + \frac{4\sigma^2}{K}, \tag{9}$$

*and*

$$\frac{1}{N} \sum_i \mathbb{E}[\|g_i^{t,K} - \nabla f_i(x^t)\|^2]$$

$$\leq 6\alpha_{\text{in}}^2 K^2 L^2 \left( \mathbb{E}[\|\nabla f(x^{t-1})\|^2] + 4L^2 X^t + 4L^2 Y^t + 4U^t + V^t \right) + \left( 24\alpha_{\text{in}}^2 K L^2 + \frac{2}{N} \right) \sigma^2. \tag{10}$$

*Proof.* To show (9), we have

$$\frac{1}{N} \sum_i \mathbb{E}[\|g_i^t - u_i^{t,0}\|^2]$$

$$\leq \frac{2}{N} \sum_i \mathbb{E}[\|u_i^{t,0} - \nabla f_i(x^t)\|^2] + \frac{2}{N} \sum_i \mathbb{E}[\|g_i^t - \nabla f_i(x^t)\|^2] \tag{11a}$$

$$\leq \frac{2}{N} \sum_i \mathbb{E}[\|u_i^{t,0} - \nabla f_i(x^t)\|^2] + 4L^2 Y^t + \frac{4\sigma^2}{K} \tag{11b}$$

$$\leq \frac{4}{N} \sum_i \mathbb{E}[\|u_i^{t,0} - \nabla f_i(x^{t-1})\|^2] + \frac{4}{N} \mathbb{E}[\|\nabla f_i(x^t) - \nabla f_i(x^{t-1})\|^2] + 4L^2 Y^t + \frac{4\sigma^2}{K} \tag{11c}$$

$$\leq \frac{4}{N} \sum_i \mathbb{E}[\|u_i^{t,0} - \nabla f_i(x^{t-1})\|^2] + 4L^2 \mathbb{E}[\|x^t - x^{t-1}\|^2] + 4L^2 Y^t + \frac{4\sigma^2}{K} \tag{11d}$$

$$= 4U^t + 4L^2 X^t + 4L^2 Y^t + \frac{4\sigma^2}{K}, \tag{11e}$$

where (11a) and (11c) use the fact $\|a+b\|^2 \leq 2\|a\|^2 + 2\|b\|^2$, (11b) uses (6) with $\theta \leftarrow 1$, and (11d) uses Assumption 1.

Lastly, we find that

$$\frac{1}{N} \sum_i \mathbb{E}[\|g_i^{t,K} - \nabla f_i(x^t)\|^2]$$

$$\leq \frac{2}{N} \sum_i \mathbb{E}[\|\nabla f_i(y_i^{t,K}) - \nabla f_i(x^t)\|^2] + \frac{2\sigma^2}{N}$$

$$\leq \frac{2L^2}{N} \sum_i \mathbb{E}[\|y_i^{t,K} - x^t\|^2] + \frac{2\sigma^2}{N}$$

$$= \frac{2\alpha_{\text{in}}^2 L^2}{N} \sum_i \mathbb{E}\Big[\Big\|\sum_k v_i^{t,k} + g_i^{t,K-1} - u_i^{t,0}\Big\|^2\Big] + \frac{2\sigma^2}{N}$$

$$= \frac{2\alpha_{\text{in}}^2 L^2}{N} \sum_i \mathbb{E}[\|K(v^t + g_i^t - u_i^{t,0})\|^2] + \frac{2\sigma^2}{N}$$

$$= \frac{2\alpha_{\text{in}}^2 K^2 L^2}{N} \sum_i \mathbb{E}[\|v^t + g_i^t - u_i^{t,0}\|^2] + \frac{2\sigma^2}{N}$$

$$\leq \frac{6\alpha_{\text{in}}^2 K^2 L^2}{N} \sum_i \big(\mathbb{E}[\|v^t - \nabla f(x^{t-1})\|^2] + \mathbb{E}[\|\nabla f(x^{t-1})\|^2] + \mathbb{E}[\|g_i^t - u_i^{t,0}\|^2]\big) + \frac{2\sigma^2}{N}$$

$$\leq 6\alpha_{\text{in}}^2 K^2 L^2 \big(V^t + \mathbb{E}[\|\nabla f(x^{t-1})\|^2] + 4U^t + 4L^2 X^t + 4L^2 Y^t + \frac{4\sigma^2}{K}\big) + \frac{2\sigma^2}{N},$$

where in the last step we apply (9). $\qquad\square$

**Lemma 7** (Descent Lemma). *Under Assumptions 1 and 2, it holds for all $t \in \mathbb{N}$ that*

$$\mathbb{E}[f(x^{t+1})] \leq \mathbb{E}[f(x^t)] - \frac{\alpha}{2} \mathbb{E}\|\nabla f(x^t)\|^2 - \Big(\frac{1}{2\alpha} - \frac{L}{2}\Big) X^{t+1} + \frac{2\alpha L^2}{S} X^t$$
$$+ \alpha L^2 \Big(1 + \frac{2}{S}\Big) Y^t + \frac{2\alpha}{S} U^t + \frac{\alpha\sigma^2}{K}\Big(\frac{2}{S} + \frac{1}{N}\Big). \tag{12}$$

*Proof.* It follows from (Li et al., 2021, Lemma 2) that

$$f(x^{t+1}) \leq f(x^t) - \frac{\alpha}{2}\|\nabla f(x^t)\|^2 - \big(\frac{1}{2\alpha} - \frac{L}{2}\big)\|x^{t+1} - x^t\|^2 + \frac{\alpha}{2}\|d^{t+1} - \nabla f(x^t)\|^2, \tag{13}$$

where recall our definition $\alpha := \alpha_{\text{in}} \alpha_{\text{out}} K$ and $d^{t+1} := \frac{1}{\alpha}(x^t - x^{t+1})$. The expectation of the last term on the right-hand side of (13) is bounded by

$$\mathbb{E}[\|d^{t+1} - \nabla f(x^t)\|^2]$$

$$= \mathbb{E}\Big[\Big\|\frac{1}{S}\sum_{i \in \mathcal{S}^t}\big(v_i^t + \frac{1}{K}(g_i^{t,K-1} - u_i^{t,0})\big) - \nabla f(x^t)\Big\|^2\Big] \tag{14a}$$

$$\leq \mathbb{E}\|g^t - \nabla f(x^t)\|^2 + \frac{1}{SN}\sum_i \mathbb{E}\|v_i^t + \frac{1}{K}(g_i^{t,K-1} - u_i^{t,0}) - v^t\|^2 \tag{14b}$$

$$\leq \frac{2L^2}{NK}\sum_{i,k}\mathbb{E}\|y_i^{t,k} - x^t\|^2 + \frac{2\sigma^2}{NK} + \frac{1}{SN}\sum_i\mathbb{E}\|v_i^t + \frac{1}{K}(g_i^{t,K-1} - u_i^{t,0}) - v^t\|^2 \tag{14c}$$

$$= \frac{2L^2}{NK}\sum_{i,k}\mathbb{E}\|y_i^{t,k} - x^t\|^2 + \frac{2\sigma^2}{NK} + \frac{1}{SN}\sum_i\mathbb{E}\|v^t + g_i^t - u_i^{t,0} - v^t\|^2 \tag{14d}$$

$$\leq 2L^2 Y^t + \frac{2\sigma^2}{NK} + \frac{1}{SN}\mathbb{E}\|g_i^t - u_i^{t,0}\|^2. \tag{14e}$$

In (14a), we apply the $x$-update (Line 17 in Algorithm 1), and in (14b), we apply Lemma 2 with $a_i \leftarrow v_i^t + \frac{1}{K}(g_i^{t,K-1} - u_i^{t,0})$, $b \leftarrow v^t$, and

$$a = \frac{1}{N}\sum_i a_i = \frac{1}{N}\sum_i \left(v_i^t + \frac{1}{K}(g_i^{t,K-1} - u_i^{t,0})\right)$$

$$= \frac{1}{N}\sum_i \left(v^t + \frac{1}{K}\left(\sum_{k=0}^{K-2} g_i^{t,k} - (K-1)u_i^{t,0}\right) + \frac{1}{K}(g_i^{t,K-1} - u_i^{t,0})\right) \quad \text{using (8)}$$

$$= \frac{1}{N}\sum_i \left(v^t - u_i^{t,0} + \frac{1}{K}\sum_k g_i^{t,k}\right)$$

$$= \frac{1}{NK}\sum_{i,k} g_i^{t,k} \equiv g^t \quad \text{using (7).}$$

Then, (14c) uses (5) with $\theta \leftarrow 1$, (14d) uses (8), and (14e) applies Cauchy–Schwarz inequality.

Combining (14e) and (9) yields

$$\mathbb{E}[\|d^{t+1} - \nabla f(x^t)\|^2] \leq \frac{4L^2}{S}X^t + 2L^2(1 + \frac{2}{S})Y^t + \frac{4}{S}U^t + \frac{2\sigma^2}{K}(\frac{2}{S} + \frac{1}{N}).$$

Substituting it back to (13) yields the desired result. $\qquad\square$

In the following three lemmas, we establish upper bounds on the quantities $U^{t+1}$, $V^{t+1}$, and $Y^t$, respectively.

**Lemma 8.** *Under Assumptions 1 and 2, it holds for all $t \in \mathbb{N}$ that*

$$U^{t+1} \leq \left(1 - \frac{S}{4N} + \frac{24\alpha_{\text{in}}^2 SK^2 L^2}{N}\right)U^t + \left(\frac{4N}{S} + \frac{24\alpha_{\text{in}}^2 SK^2 L^2}{N}\right)L^2 X^t$$

$$+ \frac{6\alpha_{\text{in}}^2 SK^2 L^2}{N}(\mathbb{E}[\|\nabla f(x^{t-1})\|^2] + 4L^2 Y^t + V^t) + \frac{2S\sigma^2}{N^2}(12\alpha_{\text{in}}^2 NKL^2 + 1). \tag{15}$$

*Proof.* It follows from the definition of the filtration $\mathcal{F}^{(k)}$, the linearity of expectation, and the update rule for $u_i^{t+1,0}$ that

$$\mathbb{E}_{\mathcal{S}}[\|u_i^{t+1,0} - \nabla f_i(x^t)\|^2 \mid \mathcal{F}^{(k)}]$$
$$= (1 - \frac{S}{N})\mathbb{E}_{\mathcal{S}}[\|u_i^{t,0} - \nabla f_i(x^t)\|^2 \mid \mathcal{F}^{(k)}] + \frac{S}{N}\mathbb{E}_{\mathcal{S}}[\|g_i^{t,K} - \nabla f_i(x^t)\|^2 \mid \mathcal{F}^{(k)}],$$

where the expectation is taken over the random sampling of clients. Then, taking the total expectation gives

$$\frac{1}{N}\sum_i \mathbb{E}[\|u_i^{t+1,0} - \nabla f_i(x^t)\|^2]$$

$$= \frac{1}{N}\sum_i \left((1 - \frac{S}{N})\mathbb{E}[\|u_i^{t,0} - \nabla f_i(x^t)\|^2] + \frac{S}{N}\mathbb{E}[\|g_i^{t,K} - \nabla f_i(x^t)\|^2]\right) \tag{16a}$$

$$\leq (1 - \frac{S}{2N})\frac{1}{N}\sum_i \mathbb{E}[\|u_i^{t,0} - \nabla f_i(x^t)\|^2] + \frac{S}{N^2}\sum_i \mathbb{E}[\|g_i^{t,K} - \nabla f_i(x^t)\|^2] \tag{16b}$$

$$\leq (1 - \frac{S}{4N})\frac{1}{N}\sum_i \mathbb{E}[\|u_i^{t,0} - \nabla f_i(x^{t-1})\|^2] + \frac{4N}{S}\frac{1}{N}\sum_i \mathbb{E}[\|\nabla f_i(x^t) - \nabla f_i(x^{t-1})\|^2]$$

$$+ \frac{S}{N^2} \sum_i \mathbb{E}[\|g_i^{t,K} - \nabla f_i(x^t)\|^2] \tag{16c}$$

$$\leq (1 - \frac{S}{4N})U^t + \frac{4NL^2}{S}X^t + \frac{S}{N^2} \sum_i \mathbb{E}[\|g_i^{t,K} - \nabla f_i(x^t)\|^2]. \tag{16d}$$

In (16a) we use Law of Total Expectation; (16b) uses the fact $S \leq N$; (16c) applies Lemma 1 with

$$\theta \leftarrow \frac{S}{4N}, \quad a \leftarrow \frac{1}{N} \sum_i \mathbb{E}[\|u_i^{t,0} - \nabla f_i(x^{t-1})\|^2], \quad b \leftarrow \frac{1}{N} \sum_i \mathbb{E}[\|\nabla f_i(x^t) - \nabla f_i(x^{t-1})\|^2];$$

and (16d) uses Assumption 1. Finally, combining (16d) and (10) yields the desired result. $\qquad\square$

**Lemma 9.** *Under Assumptions 1 and 2, it holds for all $t \in \mathbb{N}$ that*

$$V^{t+1} \leq \frac{18\alpha_{\text{in}}^2 SK^2L^2}{N^2}\left(\mathbb{E}[\|\nabla f(x^{t-1})\|^2] + 4L^2Y^t\right) + \frac{2S^3L^2}{\alpha_{\text{out}}^2 N^3}X^{t+1}$$

$$+ \left(\frac{2NL^2}{S} + \frac{3SL^2}{N^2}(24\alpha_{\text{in}}^2 K^2L^2 + 1)\right)X^t$$

$$+ \frac{3S}{N^2}(24\alpha_{\text{in}}^2 K^2L^2 + 1)U^t + \left(1 - \frac{S}{2N} + \frac{18\alpha_{\text{in}}^2 SK^2L^2}{N^2}\right)V^t$$

$$+ \left(36\alpha_{\text{in}}^2 KL^2 + \frac{3}{N} + 1\right)\frac{2S\sigma^2}{N^2}. \tag{17}$$

*Proof.* It follows from (7) and the $u$-updates in Algorithm 1 that

$$v^{t+1} = \frac{1}{N} \sum_i u_i^{t+1,0} = \frac{1}{N} \sum_i u_i^{t,0} + \frac{1}{N} \sum_{i \in \mathcal{S}^t}(g_i^{t,K} - u_i^{t,0}) = v^t + \frac{1}{N} \sum_{i \in \mathcal{S}^t}(g_i^{t,K} - u_i^{t,0}).$$

Then, it holds that

$$\mathbb{E}[\|v^{t+1} - \nabla f(x^t)\|^2]$$

$$= \mathbb{E}\left[\left\|\frac{1}{N} \sum_{i \in \mathcal{S}^t}(g_i^{t,K} - u_i^{t,0}) + v^t - \nabla f(x^t)\right\|^2\right]$$

$$\leq \mathbb{E}\left[\left\|\frac{S}{N}(g^{t,K} - v^t) + v^t - \nabla f(x^t)\right\|^2\right] + \frac{S}{N^3} \sum_i \mathbb{E}[\|g_i^{t,K} - u_i^{t,0}\|^2], \tag{18}$$

which applies Lemma 2 with $a_i = \frac{S}{N}(g_i^{t,K} - u_i^{t,0}) + v^t - \nabla f(x^t)$. Then, the first term on the right-hand side of (18) is further bounded by

$$\mathbb{E}\left[\left\|\frac{S}{N}(g^{t,K} - v^t) + v^t - \nabla f(x^t)\right\|^2\right]$$

$$= \mathbb{E}\left[\left\|(1 - \frac{S}{N})(v^t - \nabla f(x^t)) + \frac{S}{N}(g^{t,K} - \nabla f(x^t))\right\|^2\right]$$

$$\leq \left(1 - \frac{S}{N}\right)\mathbb{E}[\|v^t - \nabla f(x^t)\|^2] + \frac{S}{N}\mathbb{E}[\|g^{t,K} - \nabla f(x^t)\|^2] \tag{19a}$$

$$\leq \left(1 - \frac{S}{2N}\right)V^t + \frac{2NL^2}{S}X^t + \frac{2S}{N}\mathbb{E}\left[\left\|\frac{1}{N}\sum_i \nabla f_i(y_i^{t,K}) - \nabla f_i(x^t)\right\|^2\right] + \frac{2\sigma^2 S}{N^2} \tag{19b}$$

$$\leq \left(1 - \frac{S}{2N}\right)V^t + \frac{2NL^2}{S}X^t + \frac{2SL^2}{N^3}\mathbb{E}\left[\left\|\sum_i(y_i^{t,K} - x^t)\right\|^2\right] + \frac{2S\sigma^2}{N^2} \tag{19c}$$

$$\leq \left(1 - \frac{S}{2N}\right)V^t + \frac{2NL^2}{S}X^t + \frac{2SL^2}{N^3}\mathbb{E}\left[\left\|\sum_{i \in \mathcal{S}^t}(y_i^{t,K} - x^t)\right\|^2\right] + \frac{2S\sigma^2}{N^2} \tag{19d}$$

$$\leq \left(1 - \frac{S}{2N}\right)V^t + \frac{2NL^2}{S}X^t + \frac{2S^3L^2}{\alpha_{\text{out}}^2 N^3}X^{t+1} + \frac{2S\sigma^2}{N^2}. \tag{19e}$$

In (19a) we use Jensen's inequality. In (19b) we apply Lemma 1 with

$$\theta \leftarrow \frac{S}{2N}, \qquad a \leftarrow \mathbb{E}[\|v^t - \nabla f(x^{t-1})\|^2] \equiv V^t, \qquad b \leftarrow \mathbb{E}[\|x^t - x^{t-1}\|^2] \equiv X^t,$$

and uses Assumption 1. Then, (19c) uses again Assumption 1, and (19e) substitutes the $x$-update (Line 17 in Algorithm 1).

The second term on the right-hand side of (18) is bounded by

$$\frac{S}{N^3} \sum_i \mathbb{E}[\|g_i^{t,K} - u_i^{t,0}\|^2]$$

$$= \frac{S}{N^3} \sum_i \mathbb{E}[\|u_i^{t,0} - \nabla f_i(x^{t-1}) + \nabla f_i(x^{t-1}) - \nabla f_i(x^t) + \nabla f_i(x^t) - g_i^{t,K}\|^2]$$

$$\leq \frac{3S}{N^3} \sum_i \left(NU^t + L^2 X^t + \mathbb{E}[\|g_i^{t,K} - \nabla f_i(x^t)\|^2]\right)$$

$$\leq \frac{3S}{N^2} U^t + \frac{3SL^2}{N^2} X^t + \frac{3S}{N^3} \sum_i \mathbb{E}[\|g_i^{t,K} - \nabla f_i(x^t)\|^2]$$

$$\leq \frac{3SL^2}{N^2}(24\alpha_{\text{in}}^2 K^2 L^2 + 1)X^t + \frac{3S}{N^2}(24\alpha_{\text{in}}^2 K^2 L^2 + 1)U^t$$

$$+ \frac{18\alpha_{\text{in}}^2 SK^2 L^2}{N^2}\left(\mathbb{E}[\|\nabla f(x^{t-1})\|^2] + 4L^2 Y^t + V^t\right) + \frac{6S\sigma^2}{N^2}(12\alpha_{\text{in}}^2 KL^2 + \frac{1}{N}), \qquad (20)$$

where the last inequality uses (10). Finally, combining (18)–(20) yields the desired result. $\qquad\square$

**Lemma 10.** *Let Assumptions 1 and 2 hold, and suppose that the stepsize satisfies $\alpha_{\text{in}} KL \leq \frac{1}{2}$. Then, it holds for all $t \in \mathbb{N}$ that*

$$Y^t \leq 6e\alpha_{\text{in}}^2 K^2\left(\mathbb{E}[\|\nabla f(x^t)\|^2] + L^2 X^t + U^t\right) + e\alpha_{\text{in}}^2 K\sigma^2. \qquad (21)$$

*Proof.* When $K = 1$, $Y^t = 0$ holds trivially for all $t \in \mathbb{N}$. So we consider $K \in \mathbb{N}_{\geq 2}$ below. Using Young's inequality, we have

$$\mathbb{E}[\|y_i^{t,k+1} - x^t\|^2]$$

$$= \mathbb{E}[\|y_i^{t,k} - \alpha_{\text{in}}(g_i^{t,k} - u_i^{t,k} + v_i^{t,k}) - x^t\|^2]$$

$$= \mathbb{E}[\|y_i^{t,k} - \alpha_{\text{in}}(\nabla F(y_i^{t,k};\xi_i^{t,k}) - u_i^{t,k} + v_i^{t,k}) - x^t\|^2]$$

$$\leq \mathbb{E}[\|y_i^{t,k} - \alpha_{\text{in}}(\nabla f(y_i^{t,k}) - u_i^{t,k} + v_i^{t,k}) - x^t\|^2] + \alpha_{\text{in}}^2 \sigma^2$$

$$\leq (1 + \frac{1}{K-1})\mathbb{E}[\|y_i^{t,k} - x^t\|^2] + \alpha_{\text{in}}^2 K\mathbb{E}[\|\nabla f(y_i^{t,k}) - u_i^{t,k} + v_i^{t,k}\|^2] + \alpha_{\text{in}}^2 \sigma^2. \qquad (22)$$

The second term on the right-hand side of (22) can be further bounded using Young's inequality and Assumption 1 as

$$\alpha_{\text{in}}^2 K\mathbb{E}[\|\nabla f(y_i^{t,k}) - u_i^{t,k} + v_i^{t,k}\|^2]$$

$$= \alpha_{\text{in}}^2 K\mathbb{E}[\|\nabla f(y_i^{t,k}) - \nabla f_i(x^t) - (u_i^{t,k} - \nabla f_i(x^t)) + v_i^{t,k} - \nabla f(x^t) + \nabla f(x^t)\|^2]$$

$$\leq 3\alpha_{\text{in}}^2 KL^2\mathbb{E}[\|y_i^{t,k} - x^t\|^2] + 3\alpha_{\text{in}}^2 K\mathbb{E}[\|u_i^{t,k} - \nabla f_i(x^t) - v_i^{t,k} + \nabla f(x^t)\|^2]$$

$$+ 3\alpha_{\text{in}}^2 K\mathbb{E}[\|\nabla f(x^t)\|^2]. \qquad (23)$$

The second term on the right-hand side of (23) can be bounded using Young's equality and Assumption 1

$$\frac{3\alpha_{\text{in}}^2 K}{N} \sum_i \mathbb{E}[\|u_i^{t,k} - \nabla f_i(x^t) - v_i^{t,k} + \nabla f(x^t)\|^2]$$

$$= \frac{3\alpha_{\text{in}}^2 K}{N} \sum_i \mathbb{E}[\|u_i^{t,0} - \nabla f_i(x^t) - v_i^{t,0} + \nabla f(x^t)\|^2] \qquad (24a)$$

$$\leq \frac{3\alpha_{\text{in}}^2 K}{N} \sum_i \mathbb{E}[\|u_i^{t,0} - \nabla f_i(x^t)\|^2] \qquad (24b)$$

$$\leq \frac{6\alpha_{\text{in}}^2 K}{N} \sum_i \left(\mathbb{E}[\|u_i^{t,0} - \nabla f_i(x^{t-1})\|^2] + L^2\mathbb{E}[\|x^t - x^{t-1}\|^2]\right), \qquad (24c)$$

where (24a) uses Line 9 in Algorithm 1:

$$v_i^{t,k+1} = v_i^{t,k} + u_i^{t,k+1} - u_i^{t,k}$$

$$u_i^{t,k+1} - v_i^{t,k+1} = u_i^{t,k} - v_i^{t,k} = \cdots = u_i^{t,0} - v_i^{t,0},$$

(24b) uses (7), and (24c) uses Young's equality and Assumption 1.

Then, combining (22), (23), and (24c) yields

$$\frac{1}{N} \sum_i \mathbb{E}[\|y_i^{t,k+1} - x^t\|^2]$$

$$\leq \frac{1}{N} \sum_i \left( \frac{K}{K-1} \mathbb{E}[\|y_i^{t,k} - x^t\|^2] + \alpha_{\text{in}}^2 K \mathbb{E}[\|\nabla f_i(y_i^{t,k}) - u_i^t - v_i^{t,k}\|^2] + \alpha_{\text{in}}^2 \sigma^2 \right)$$

$$\leq \frac{1}{N} \sum_i \left( \frac{K}{K-1} \mathbb{E}[\|y_i^{t,k} - x^t\|^2] + 3\alpha_{\text{in}}^2 L^2 K \mathbb{E}[\|y_i^{t,k} - x^t\|^2] + 3\alpha_{\text{in}}^2 K \mathbb{E}[\|\nabla f(x^t)\|^2] \right)$$

$$+ \frac{1}{N} \sum_i \left( 3\alpha_{\text{in}}^2 K \mathbb{E}[\|u_i^{t,k} - \nabla f_i(x^t) - v_i^{t,k} + \nabla f(x^t)\|^2] \right) + \alpha_{\text{in}}^2 \sigma^2$$

$$\leq \frac{6\alpha_{\text{in}}^2 K}{N} \sum_i \left( \mathbb{E}[\|u_i^{t,0} - \nabla f_i(x^{t-1})\|^2] + L^2 \mathbb{E}[\|x^t - x^{t-1}\|^2] + \mathbb{E}[\|\nabla f(x^t)\|^2] \right)$$

$$+ \frac{1}{N} \sum_i \left( \frac{K}{K-1} + 3\alpha_{\text{in}}^2 K L^2 \right) \mathbb{E}[\|y_i^{t,k} - x^t\|^2] + \alpha_{\text{in}}^2 \sigma^2$$

$$\leq \alpha_{\text{in}}^2 \sum_{\ell=0}^k \left( \frac{K}{K-1} + 3\alpha_{\text{in}}^2 K L^2 \right)^\ell \left( K \left( \mathbb{E}[\|\nabla f(x^t)\|^2] + L^2 X^t + U^t \right) + \sigma^2 \right) \tag{25a}$$

$$\leq \alpha_{\text{in}}^2 \sum_{\ell=0}^k \left( \frac{K+1}{K-1} \right)^\ell \left( 6K \left( \mathbb{E}[\|\nabla f(x^t)\|^2] + L^2 X^t + U^t \right) + \sigma^2 \right). \tag{25b}$$

In (25a), we apply the fact that

$$\phi^{k+1} \leq \theta \phi^k + \nu \leq \theta(\theta \phi^{k-1} + \nu) + \nu \leq \cdots \leq \theta^{k+1} \phi^0 + \nu \sum_{\ell=0}^k \theta^\ell$$

to

$$\phi^k \leftarrow \mathbb{E}[\|y_i^{t,k} - x^t\|^2\|], \qquad \theta \leftarrow \frac{K}{K-1} + 3\alpha_{\text{in}}^2 K L^2,$$

$$\nu \leftarrow 6\alpha_{\text{in}}^2 K \left( \mathbb{E}[\|\nabla f(x^t)\|^2] + L^2 X^t + U^t \right) + \alpha_{\text{in}}^2 \sigma^2,$$

and recall the initialization $y_i^{t,0} = x^t$ for all $i \in [N]$. Then, in (25b) we use the stepsize condition $\alpha_{\text{in}} K L \leq \frac{1}{2}$ so that $3\alpha_{\text{in}}^2 K L^2 \leq \frac{3}{4K} \leq \frac{1}{K-1}$.

Finally, iterating and averaging (25b) over $k = 0, \ldots, K-1$, we obtain

$$Y^t \leq \frac{\alpha_{\text{in}}^2}{K} \sum_k \sum_{\ell=0}^{k-1} \left( \frac{K+1}{K-1} \right)^\ell \left( 6K(\mathbb{E}[\|\nabla f(x^t)\|^2] + L^2 X^t + U^t) + \sigma^2 \right)$$

$$\leq \alpha_{\text{in}}^2 K \left( 1 + \frac{1}{K-1} \right)^{K-1} (6K(\mathbb{E}[\|\nabla f(x^t)\|^2] + L^2 X^t + U^t) + \sigma^2)$$

$$\leq 6e\alpha_{\text{in}}^2 K^2 (\mathbb{E}[\|\nabla f(x^t)\|^2] + L^2 X^t + U^t) + e\alpha_{\text{in}}^2 K \sigma^2,$$

where in the second inequality we relax $(\frac{K+1}{K-1})^\ell$ to $(\frac{K+1}{K-1})^{K-1}$, and the last inequality uses the fact $(1 + \frac{1}{K-1})^{K-1} \leq e$ for all $K \geq 2$. $\qquad \square$

**Theorem 3** (Restatement of Theorem 1). *Suppose Assumptions 1 and 2 hold, and suppose the stepsizes satisfy $\alpha = \alpha_{\text{in}} \alpha_{\text{out}} K$, $144\alpha_{\text{in}}^2 K^2 L^2 \leq \alpha$, and*

$$\alpha \leq \min \left\{ \frac{2}{9}, \frac{1}{L}, \left( \frac{KLR_1}{4ST\sigma^2} \right)^{1/3}, \left( \frac{N^2 LR_1}{33SK\sigma^2} \right)^{1/2} \right\}.$$

*Then, the $\{x^t\}$ sequence generated by Algorithm 1 satisfies*

$$\frac{1}{T} \sum_{t=0}^{T-1} \mathbb{E}[\|\nabla f(x^t)\|^2] \leq \gamma_1 \sqrt{\frac{LR_1\sigma^2}{SKT}} + \frac{L(\gamma_2 R_1 + \gamma_3 R_2)}{T} \left( \frac{N}{S} \right)^{2/3},$$

*where $R_1 := f(x^0) - \inf_x f(x)$, $R_2 := \mathbb{E}[\|\nabla f(x^0)\|^2]$, and $(\gamma_1, \gamma_2, \gamma_3) \in \mathbb{R}^3_{++}$ are numeric constants.*

*Proof.* Combining the stepsize conditions $144\alpha_{\text{in}}^2 K^2 L^2 \leq \alpha$ and $\alpha \leq 1$ with (12), (15), (17), and (21) yields

$$\mathbb{E}[f(x^{t+1})] \leq \mathbb{E}[f(x^t)] - \tfrac{\alpha}{2}\mathbb{E}\|\nabla f(x^t)\|^2 - \left(\tfrac{1}{2\alpha} - \tfrac{L}{2}\right)X^{t+1} + \tfrac{2\alpha L^2}{S}X^t$$
$$+ 3\alpha L^2 Y^t + \tfrac{2\alpha}{S}U^t + \tfrac{\alpha\sigma^2}{K}\left(\tfrac{2}{S} + \tfrac{1}{N}\right) \tag{26}$$

$$U^{t+1} \leq \left(1 - \tfrac{\alpha S}{12N}\right)U^t + \left(\tfrac{4N}{S} + \tfrac{\alpha S}{6N}\right)L^2 X^t$$
$$+ \tfrac{\alpha S}{24N}\left(\mathbb{E}[\|\nabla f(x^{t-1})\|^2] + 4L^2 Y^t + V^t\right) + \tfrac{S(N+12K)\sigma^2}{6N^2 K} \tag{27}$$

$$V^{t+1} \leq \tfrac{\alpha S}{8N^2}\left(\mathbb{E}[\|\nabla f(x^{t-1})\|^2] + 4L^2 Y^t\right) + \tfrac{S^3}{36N^3}X^{t+1} + \left(\tfrac{2N}{S} + \tfrac{4S}{N^2}\right)L^2 X^t$$
$$+ \tfrac{4S}{N^2}U^t + \left(1 - \tfrac{\alpha S}{4N}\right)V^t + \left(\tfrac{1}{4K} + \tfrac{3}{N} + 1\right)\tfrac{2S\sigma^2}{N^2} \tag{28}$$

$$Y^t \leq \tfrac{e\alpha}{24L^2}\left(\mathbb{E}[\|\nabla f(x^t)\|^2] + L^2 X^t + U^t\right) + \tfrac{e\alpha\sigma^2}{144KL^2}. \tag{29}$$

Adding together (26), $\tfrac{12\alpha N}{S} \times$ (27), $\tfrac{2\alpha N}{S} \times$ (28), and $\tfrac{4\alpha L^2}{e} \times$ (29) gives

$$\mathbb{E}[f(x^{t+1})] + \tfrac{12\alpha N}{S}U^{t+1} + \tfrac{2\alpha N}{S}V^{t+1} + \tfrac{4\alpha L^2}{e}Y^t$$
$$\leq \mathbb{E}[f(x^t)] - \tfrac{\alpha}{3}\mathbb{E}[\|\nabla f(x^t)\|^2] + \tfrac{3\alpha^2 N}{8S}\mathbb{E}[\|\nabla f(x^{t-1})\|^2] - \left(\tfrac{1}{2\alpha} - \tfrac{L}{2} - \tfrac{\alpha S^2}{18N^2}\right)(X^{t+1} - X^t)$$
$$+ \tfrac{12\alpha N}{S}U^t + \tfrac{2\alpha N}{S}V^t + \tfrac{4\alpha L^2}{e}Y^t + \alpha\sigma^2\left(\tfrac{73}{36K} + \tfrac{2}{SK} + \tfrac{32}{N} + \tfrac{1}{NK}\right) \tag{30a}$$
$$\leq \mathbb{E}[f(x^t)] - \tfrac{\alpha}{4}\mathbb{E}[\|\nabla f(x^t)\|^2] - \tfrac{\alpha}{12}\left(\mathbb{E}[\|\nabla f(x^t)\|^2] - \mathbb{E}[\|\nabla f(x^{t-1})\|^2]\right)$$
$$- \left(\tfrac{1}{2\alpha} - \tfrac{L}{2} - \tfrac{\alpha S^2}{18N^2}\right)(X^{t+1} - X^t) + \tfrac{12\alpha S}{N}U^t + \tfrac{2\alpha S}{N}V^t + \tfrac{4\alpha L^2}{e}Y^t$$
$$+ \alpha\sigma^2\left(\tfrac{73}{36K} + \tfrac{2}{SK} + \tfrac{32}{N} + \tfrac{1}{NK}\right), \tag{30b}$$

where in (30a) we use the fact that $N \geq S \geq 1$, and (30b) uses the stepsize condition $\alpha \leq \tfrac{2}{9}$. In addition, the stepsize condition $\alpha \leq \min\{\tfrac{2}{9}, \tfrac{1}{L}\}$ guarantees that $\tfrac{1}{2\alpha} - \tfrac{L}{2} - \tfrac{S^3}{36N^3} \geq 0$.

Now define the Lyapunov function for all $t \in \mathbb{N}$,

$$\Phi^t := \mathbb{E}[f(x^t)] - f^\star + \tfrac{\alpha}{12}\mathbb{E}[\|\nabla f(x^{t-1})\|^2] + \left(\tfrac{1}{2\alpha} - \tfrac{L}{2} - \tfrac{\alpha S^3}{36N^3}\right)X^t + \tfrac{12\alpha N}{S}U^t + \tfrac{2\alpha N}{S}V^t,$$

where $f^\star := \inf_x f(x)$. With the convention $x^{-1} = x^0$, the initial condition reduces to

$$\Phi^0 = f(x^0) - f^\star + \tfrac{\alpha}{12}\mathbb{E}[\|\nabla f(x^0)\|^2] + \tfrac{12\alpha N}{S}\sum_{i=1}^{N}\|u_i^{0,0} - \nabla f_i(x^0)\|^2 + \tfrac{2\alpha N}{S}\|v^0 - \nabla f(x^0)\|^2$$

$$= R_1 + \tfrac{\alpha R_2}{12}.$$

Substituting the definition of $\Phi^t$ into (30b) gives

$$\tfrac{\alpha}{4}\mathbb{E}[\|\nabla f(x^t)\|^2] \leq \Phi^t - \Phi^{t+1} + \alpha\sigma^2\left(\tfrac{73}{36K} + \tfrac{2}{SK} + \tfrac{32}{N} + \tfrac{1}{NK}\right)$$

$$\tfrac{1}{T}\sum_{t=0}^{T-1}\mathbb{E}[\|\nabla f(x^t)\|^2] \leq \tfrac{4(\Phi^0 - \Phi^T)}{\alpha T} + \alpha\sigma^2\left(\tfrac{73}{36K} + \tfrac{2}{SK} + \tfrac{32}{N} + \tfrac{1}{NK}\right)$$

$$\leq \tfrac{4R_1}{\alpha T} + \tfrac{\alpha R_2}{3T} + \alpha\sigma^2\left(\tfrac{73}{36K} + \tfrac{2}{SK} + \tfrac{32}{N} + \tfrac{1}{NK}\right).$$

Finally, plugging the condition on $\alpha$ completes the proof. $\qquad\square$

## D  CONVERGENCE ANALYSIS FOR ALGORITHM 2

Similar to the convention in §B.1, we denote $\alpha := \alpha_{\text{in}}\alpha_{\text{out}}K$ and

$$\tilde{d}^{t+1} := \tfrac{1}{\alpha}(x^t - x^{t+1}) \equiv \tfrac{1}{S}\sum_{i \in \mathcal{S}^t}\mathcal{C}_i\left(\beta_1\left(v_i^t + \tfrac{1}{K}(g_i^{t,K-1} - u_i^{t,0})\right)\right),$$

$$d^{t+1} := \frac{\beta_1}{S}\big(v_i^t + \frac{1}{K}(g_i^{t,K-1} - u_i^{t,0})\big),$$

$$u_i^{t+1,0} = \begin{cases} u_i^{t,0} + \mathcal{C}_i\big(\beta_1(g_i^{t,K} - u_i^{t,0})\big) & \text{if } i \in \mathcal{S}^t \\ u_i^{t,0} & \text{otherwise.} \end{cases}$$

It is apparent that Lemma 5, Lemma 6, and Lemma 10 still hold for Algorithm 2.

**Lemma 11.** *Under Assumptions 1 and 2, it holds for all $t \in \mathbb{N}$ that*

$$\frac{1}{N}\sum_i \mathbb{E}[\|g_i^{t,K} - u_i^{t,0}\|^2]$$

$$\leq 3L^2(24\alpha_{\text{in}}^2 K^2 L^2 + 1)X^t + 3(24\alpha_{\text{in}}^2 K^2 L^2 + 1)U^t$$

$$+ 18\alpha_{\text{in}}^2 K^2 L^2 \big(\mathbb{E}[\|\nabla f(x^{t-1})\|^2] + 4L^2 Y^t + V^t\big) + 6\left(12\alpha_{\text{in}}^2 K L^2 + \frac{1}{N}\right)\sigma^2. \tag{31}$$

*Proof.* It holds that

$$\frac{1}{N}\sum_i \mathbb{E}[\|g_i^{t,K} - u_i^{t,0}\|^2]$$

$$= \frac{1}{N}\sum_i \mathbb{E}[\|u_i^{t,0} - \nabla f_i(x^{t-1}) + \nabla f_i(x^{t-1}) - \nabla f_i(x^t) + \nabla f_i(x^t) - g_i^{t,K}\|^2]$$

$$\leq \frac{3}{N}\sum_i \big(\mathbb{E}[\|u_i^{t,0} - \nabla f_i(x^{t-1})\|^2] + L^2 \mathbb{E}[\|x^t - x^{t-1}\|^2] + \mathbb{E}[\|g_i^{t,K} - \nabla f_i(x^t)\|^2]\big)$$

$$\leq \frac{3}{N}\sum_i \mathbb{E}[\|u_i^{t,0} - \nabla f_i(x^{t-1})\|^2] + 3L^2 \mathbb{E}[\|x^t - x^{t-1}\|^2] + \frac{3}{N}\sum_i \mathbb{E}[\|g_i^{t,K} - \nabla f_i(x^t)\|^2]$$

$$\leq 3L^2(24\alpha_{\text{in}}^2 K^2 L^2 + 1)\mathbb{E}[\|x^t - x^{t-1}\|^2] + 3(24\alpha_{\text{in}}^2 K^2 L^2 + 1)\frac{1}{N}\sum_i \mathbb{E}[\|u_i^{t,0} - \nabla f_i(x^{t-1})\|^2]$$

$$+ 18\alpha_{\text{in}}^2 K^2 L^2 \big(\mathbb{E}[\|\nabla f(x^{t-1})\|^2] + 4L^2 Y^t + \mathbb{E}[\|v^t - \nabla f(x^{t-1})\|^2]\big) + 6\sigma^2(12\alpha_{\text{in}}^2 K L^2 + \frac{1}{N}),$$

where in the last step we use (10). $\qquad\square$

**Lemma 12** (Descent Lemma). *Under Assumptions 1 and 2, it holds for all $t \in \mathbb{N}$ and $(\alpha, \beta_1, \beta_2) \in \mathbb{R}_{++}^3$ that*

$$\mathbb{E}[f(x^{t+1})]$$

$$\leq \mathbb{E}[f(x^t)] - \left(\frac{\alpha}{2} - 2(1-\beta_1)^2\right)\mathbb{E}[\|\nabla f(x^t)\|^2] + \frac{3\alpha\beta_1^2 \omega N}{S}\mathbb{E}[\|\nabla f(x^{t-1})\|^2]$$

$$- \left(\frac{1}{2\alpha} - \frac{L}{2}\right)X^{t+1} + \frac{4\alpha\beta_1^2(1+\omega)L^2}{S}X^t + 4\alpha\beta_1^2\left(1 + \frac{1+\omega}{S}\right)L^2 Y^t + \frac{4\alpha\beta_1^2(1+\omega)}{S}U^t$$

$$+ \frac{3\alpha\beta_1^2 \omega N}{S}V^t + \frac{4\alpha\beta_1^2 \sigma^2}{K}\left(\frac{1+\omega}{S} + \frac{1}{N}\right). \tag{32}$$

*Proof.* It follows from (Li et al., 2021, Lemma 2) that

$$f(x^{t+1}) \leq f(x^t) - \frac{\alpha}{2}\|\nabla f(x^t)\|^2 - \left(\frac{1}{2\alpha} - \frac{L}{2}\right)\|x^{t+1} - x^t\|^2 + \frac{\alpha}{2}\|\tilde{d}^{t+1} - \nabla f(x^t)\|^2$$

$$\leq f(x^t) - \frac{\alpha}{2}\|\nabla f(x^t)\|^2 - \left(\frac{1}{2\alpha} - \frac{L}{2}\right)\|x^{t+1} - x^t\|^2$$

$$+ \alpha\|\tilde{d}^{t+1} - d^{t+1}\|^2 + \alpha\|d^{t+1} - \nabla f(x^t)\|^2, \tag{33}$$

where recall our definition $\alpha := \alpha_{\text{in}}\alpha_{\text{out}}K$ and $\tilde{d}^{t+1} := \frac{1}{\alpha}(x^t - x^{t+1})$. The expectation of the last term on the right-hand side of (33) is bounded by

$$\mathbb{E}[\|d^{t+1} - \nabla f(x^t)\|^2]$$

$$= \mathbb{E}\left[\left\|\frac{\beta_1}{S}\sum_{i \in \mathcal{S}^t}(v_i^t + \frac{1}{K}(g_i^{t,K-1} - u_i^{t,0})) - \nabla f(x^t)\right\|^2\right] \tag{34a}$$

$$\leq \mathbb{E}\|\beta_1 g^t - \nabla f(x^t)\|^2 + \tfrac{\beta_1^2}{SN}\sum_i \mathbb{E}\|v_i^t + \tfrac{1}{K}(g_i^{t,K-1} - u_i^{t,0}) - v^t\|^2 \tag{34b}$$

$$\leq \mathbb{E}\|\beta_1(g^t - \nabla f(x^t)) - (1-\beta_1)\nabla f(x^t)\|^2 + \tfrac{\beta_1^2}{SN}\sum_i \mathbb{E}\|v_i^t + \tfrac{1}{K}(g_i^{t,K-1} - u_i^{t,0}) - v^t\|^2 \tag{34c}$$

$$\leq 2(1-\beta_1)^2\mathbb{E}\|\nabla f(x^t)\|^2 + 2\beta_1^2\mathbb{E}\|g^t - \nabla f(x^t)\|^2 + \tfrac{\beta_1^2}{SN}\sum_i \mathbb{E}\|v_i^t + \tfrac{1}{K}(g_i^{t,K-1} - u_i^{t,0}) - v^t\|^2 \tag{34d}$$

$$\leq 2(1-\beta_1)^2\mathbb{E}\|\nabla f(x^t)\|^2 + 4\beta_1^2 L^2 Y^t + \tfrac{4\beta_1^2\sigma^2}{NK} + \tfrac{\beta_1^2}{SN}\sum_i \mathbb{E}[\|g_i^t - u_i^{t,0}\|^2] \tag{34e}$$

$$\leq 2(1-\beta_1)^2\mathbb{E}\|\nabla f(x^t)\|^2 + 4\beta_1^2 L^2 Y^t + \tfrac{4\beta_1^2}{S}U^t + \tfrac{4\beta_1^2 L^2}{S}X^t + \tfrac{4\beta_1^2 L^2}{S}Y^t + \tfrac{4\beta_1^2\sigma^2}{K}(\tfrac{1}{S} + \tfrac{1}{N}). \tag{34f}$$

In (34a), we apply the $x$-update (Line 17 in Algorithm 2), and in (34b), we apply Lemma 2. Then, (34e) uses (5) with $\theta \leftarrow 1$ and (8), and finally, (34f) uses (9).

The second-to-last term on the right-hand side of (33) is bounded by

$$\mathbb{E}[\|\tilde{d}^{t+1} - d^{t+1}\|^2]$$

$$\leq \tfrac{\beta_1^2\omega}{S^2}\mathbb{E}\Big[\sum_{i\in\mathcal{S}^t}\|v_i^t + \tfrac{1}{K}(g_i^{t,K-1} - u_i^{t,0})\|^2\Big] \tag{35a}$$

$$\leq \tfrac{\beta_1^2\omega}{SN}\sum_i \mathbb{E}[\|v_i^t + \tfrac{1}{K}(g_i^{t,K-1} - u_i^{t,0})\|^2] \tag{35b}$$

$$= \tfrac{\beta_1^2\omega}{SN}\sum_i \mathbb{E}[\|v^t + g_i^t - u_i^{t,0}\|^2] \tag{35c}$$

$$\leq \tfrac{3\beta_1^2\omega}{SN}\sum_i \big(\mathbb{E}[\|v^t - \nabla f(x^{t-1})\|^2] + \mathbb{E}[\|\nabla f(x^{t-1})\|^2] + \mathbb{E}[\|g_i^t - u_i^{t,0}\|^2]\big) \tag{35d}$$

$$\leq \tfrac{\beta_1^2\omega}{S}\big(V^t + \mathbb{E}[\|\nabla f(x^{t-1})\|^2]\big) + \tfrac{4\beta_1^2\omega L^2}{S}X^t + \tfrac{4\beta_1\omega L^2}{S}Y^t + \tfrac{4\beta_1^2\omega}{S}U^t + \tfrac{4\beta_1^2\omega\sigma^2}{SK}. \tag{35e}$$

In (35a), we use the definition of $d^{t+1}$ and $\tilde{d}^{t+1}$, and (35c) uses (8). In (35d), we use Young's inequality, and (35e) uses (9).

Finally, combining (33), (34f), and (35e) yields the desired result. $\qquad\square$

**Lemma 13.** *Under Assumptions 1 and 2, it holds for all $t \in \mathbb{N}$ that*

$$U^{t+1} \leq \left(1 - \frac{S}{4N} + \frac{4(1-\beta_2)^2 S}{N} + \frac{3\omega\beta_2^2 S}{N}(24\alpha_{\text{in}}^2 K^2 L^2 + 1) + \frac{48\alpha_{\text{in}}^2\beta_2^2 SK^2 L^2}{N}\right)U^t$$

$$+ \left(\frac{4N}{S} + \frac{4(1-\beta_2)^2 S}{N} + \frac{3\omega\beta_2^2 S}{N}(24\alpha_{\text{in}}^2 K^2 L^2 + 1) + \frac{48\alpha_{\text{in}}^2\beta_2^2 SK^2 L^2}{N}\right)L^2 X^t$$

$$+ \left(\frac{12\alpha_{\text{in}}^2\beta_2^2 SK^2 L^2}{N} + \frac{18\omega\beta_2^2\alpha_{\text{in}}^2 SK^2 L^2}{N}\right)\left(\mathbb{E}[\|\nabla f(x^{t-1})\|^2] + 4L^2 Y^t + V^t\right)$$

$$+ \frac{2\beta_2^2 S\sigma^2}{N}\left(12(2+3\omega)\alpha_{\text{in}}^2 KL^2 + \frac{2+3\omega}{N}\right). \tag{36}$$

*Proof.* It follows from the definition of the filtration $\mathcal{F}^{(k)}$, the linearity of expectation, and the update rule for $u_i^{t+1,0}$ that

$$\mathbb{E}_{\mathcal{S}}[\|u_i^{t+1,0} - \nabla f_i(x^t)\|^2 \mid \mathcal{F}^{(k)}] = (1 - \tfrac{S}{N})\mathbb{E}_{\mathcal{S}}[\|u_i^{t,0} - \nabla f_i(x^t)\|^2 \mid \mathcal{F}^{(k)}]$$
$$+ \tfrac{S}{N}\mathbb{E}_{\mathcal{S}}[\|u_i^{t,0} + \mathcal{C}_i(\beta_2(g_i^{t,K} - u_i^{t,0})) - \nabla f_i(x^t)\|^2 \mid \mathcal{F}^{(k)}],$$

where the expectation is taken over the random sampling of clients. Then, taking similar steps as in the proof of Lemma 8 gives

$$U^{t+1}$$

$$= \frac{1}{N} \sum_i \left( (1 - \frac{S}{N}) \mathbb{E}[\|u_i^{t,0} - \nabla f_i(x^t)\|^2] + \frac{S}{N} \mathbb{E}[\|u_i^{t,0} + \mathcal{C}_i(\beta_2(g_i^{t,K} - u_i^{t,0})) - \nabla f_i(x^t)\|^2] \right)$$

$$\leq (1 - \frac{S}{2N}) \frac{1}{N} \sum_i \mathbb{E}[\|u_i^{t,0} - \nabla f_i(x^t)\|^2] + \frac{S}{N^2} \sum_i \mathbb{E}[\|u_i^{t,0} + \mathcal{C}_i(\beta_2(g_i^{t,K} - u_i^{t,0})) - \nabla f_i(x^t)\|^2]$$

$$\leq (1 - \frac{S}{4N}) \frac{1}{N} \sum_i \mathbb{E}[\|u_i^{t,0} - \nabla f_i(x^{t-1})\|^2] + \frac{4N}{S} \frac{1}{N} \sum_i \mathbb{E}[\|\nabla f_i(x^t) - \nabla f_i(x^{t-1})\|^2]$$

$$+ \frac{S}{N^2} \sum_i \mathbb{E}[\|u_i^{t,0} + \beta_2(g_i^{t,K} - u_i^{t,0}) - \nabla f_i(x^t)\|^2] + \frac{S\omega\beta_2^2}{N^2} \sum_i \mathbb{E}[\|g_i^{t,K} - u_i^{t,0}\|^2]$$

$$\leq (1 - \frac{S}{4N}) \frac{1}{N} \sum_i \mathbb{E}[\|u_i^{t,0} - \nabla f_i(x^{t-1})\|^2] + \frac{4NL^2}{S} \mathbb{E}[\|x^t - x^{t-1}\|^2]$$

$$+ \frac{S}{N^2} \sum_i \mathbb{E}[\|u_i^{t,0} + \beta_2(g_i^{t,K} - u_i^{t,0}) - \nabla f_i(x^t)\|^2] + \frac{\omega\beta_2^2 S}{N^2} \sum_i \mathbb{E}[\|g_i^{t,K} - u_i^{t,0}\|^2]. \tag{37}$$

Then, the first term on the right-hand of (37) can be further bounded as

$$\frac{S}{N^2} \sum_i \mathbb{E}[\|u_i^{t,0} + \beta_2(g_i^{t,K} - u_i^{t,0}) - \nabla f_i(x^t)\|^2]$$

$$= \frac{S}{N^2} \sum_i \mathbb{E}[\|\beta_2(g_i^{t,K} - \nabla f_i(x^t)) + (1 - \beta_2)(u_i^{t,0} - \nabla f_i(x^t))\|^2]$$

$$\leq \frac{2(1-\beta_2)^2 S}{N^2} \sum_i \mathbb{E}[\|(u_i^{t,0} - \nabla f_i(x^t))\|^2] + \frac{2\beta_2^2 S}{N^2} \sum_i \mathbb{E}[\|(g_i^{t,K} - \nabla f_i(x^t))\|^2]$$

$$\leq \frac{4(1-\beta_2)^2 S}{N} U^t + \frac{4(1-\beta_2)^2 S L^2}{N} X^t + \frac{2\beta_2^2 S}{N^2} \sum_i \mathbb{E}[\|g_i^{t,K} - \nabla f_i(x^t)\|^2]. \tag{38}$$

Finally, combining (37), (38) and Lemma 11 yields the desired result. $\qquad\square$

**Lemma 14.** *Under Assumptions 1 and 2, it holds for all $t \in \mathbb{N}$ that*

$$V^{t+1} \leq \left( 1 - \frac{\beta_2 S}{2N} + \frac{18(1+\omega)\beta_2^2 L^2 \alpha_{\text{in}}^2 K^2 S}{N^2} \right) V^t + \frac{3(1+\omega)\beta_2^2 S(1 + 24\alpha_{\text{in}}^2 K^2 L^2)}{N^2} U^t$$

$$+ \frac{18(1+\omega)\beta_2^2 \alpha_{\text{in}}^2 SK^2 L^2}{N^2} \left( \mathbb{E}[\|\nabla f(x^{t-1})\|^2] + 4L^2 Y^t \right)$$

$$+ \left( \frac{2NL^2}{S\beta_2} + \frac{3(1+\omega)\beta_2^2 SL^2}{N^2}(24\alpha_{\text{in}}^2 K^2 L^2 + 1) \right) X^t + \frac{2\beta_2 S^3 L^2}{\alpha_{\text{out}}^2 N^3} X^{t+1}$$

$$+ \left( 36(1+\omega)\beta_2^2 \alpha_{\text{in}}^2 KL^2 + \frac{3(1+\omega)\beta_2^2}{N} + \beta_2 \right) \frac{2S\sigma^2}{N^2}. \tag{39}$$

*Proof.* It follows from Lemma 5 that

$$v^{t+1} = \frac{1}{N} \sum_i u_i^{t+1,0} = \frac{1}{N} \left( \sum_i u_i^{t,0} + \sum_{i \in \mathcal{S}^t} \mathcal{C}_i(\beta_2(g_i^{t,K} - u_i^{t,0})) \right)$$

$$= v^t + \frac{1}{N} \sum_{i \in \mathcal{S}^t} \mathcal{C}_i\left(\beta_2(g_i^{t,K} - u_i^{t,0})\right).$$

Then, we have

$$V^{t+1} = \mathbb{E}\left[ \left\| \frac{1}{N} \sum_{i \in \mathcal{S}^t} \mathcal{C}_i\left(\beta_2(g_i^{t,K} - u_i^{t,0})\right) + v^t - \nabla f(x^t) \right\|^2 \right]$$

$$\leq \mathbb{E}\left[ \left\| \frac{1}{N} \sum_{i \in \mathcal{S}^t} \beta_2(g_i^{t,K} - u_i^{t,0}) + v^t - \nabla f(x^t) \right\|^2 \right] + \frac{\omega\beta_2^2}{N^2} \sum_{i \in \mathcal{S}^t} \mathbb{E}[\|g_i^{t,K} - u_i^{t,0}\|^2]$$

$$\leq \mathbb{E}\left[ \left\| \frac{S\beta_2}{N}(g^{t,K} - v^t) + v^t - \nabla f(x^t) \right\|^2 \right] + \frac{(1+\omega)\beta_2^2 S}{N^2} \frac{1}{N} \sum_i \mathbb{E}[\|g_i^{t,K} - u_i^{t,0}\|^2],$$

where $g^{t,K} := \frac{1}{N} \sum_i g_i^{t,K}$. Then, combining it with Lemma 11 yields the desired result. $\qquad\square$

**Theorem 4** (Restatement of Theorem 2). *Suppose Assumptions 1 and 2 hold, and suppose the stepsizes satisfy $\alpha = \alpha_{in}\alpha_{out}K$, $144\alpha_{in}^2 K^2 L^2 \le \alpha$, $\beta_1 = \frac{1-\alpha}{4}$, $\beta_2 = \frac{\alpha SL}{27N}$, and*

$$\alpha \le \min\left\{\frac{2}{9(1+\omega)}, \frac{1}{L}, \left(\frac{KLR_1}{4(1+\omega)ST\sigma^2}\right)^{1/3}, \left(\frac{N^2LR_1}{50(1+\omega)SK\sigma^2}\right)^{1/2}\right\}.$$

*Then, the $\{x^t\}$ sequence generated by Algorithm 2 satisfies*

$$\frac{1}{T}\sum_{t=0}^{T-1}\mathbb{E}[\|\nabla f(x^t)\|^2]$$

$$\le \gamma_1\sqrt{\frac{(1+\omega)LR_1\sigma^2}{SKT}} + \frac{(1+\omega)L(\gamma_2 R_1 + \gamma_3 R_2)}{T}\left(\frac{N}{S}\right)^{1/3} + \gamma_4\left(\frac{(1+\omega)N^2L^2R_1^2\sigma^2}{S^3KT^2}\right)^{1/3},$$

*where $R_1 := f(x^0) - \inf_x f(x)$, $R_2 := \mathbb{E}[\|\nabla f(x^0)\|^2]$, and $(\gamma_1, \gamma_2, \gamma_3, \gamma_4) \in \mathbb{R}_{++}^4$ are numeric constants.*

*Proof.* The key idea in the proof is similar to the proof of Theorem 1. Adding together (32), $\frac{12\alpha N}{S}\times$ (36), $\frac{4\alpha N}{S}\times$ (39), and $\frac{4\alpha L^2}{e}\times$ (21) gives

$$\mathbb{E}[f(x^{t+1})] + \frac{12\alpha N}{S}U^{t+1} + \frac{4\alpha N}{S}V^{t+1} + \frac{4\alpha L^2}{e}Y^t$$
$$\le \mathbb{E}[f(x^t)] - \frac{\alpha}{4}\mathbb{E}[\|\nabla f(x^t)\|^2] - \frac{\alpha}{12}\left(\mathbb{E}[\|\nabla f(x^t)\|^2] - \mathbb{E}[\|\nabla f(x^{t-1})\|^2]\right)$$
$$- \left(\frac{1}{2(1+\omega)\alpha} - \frac{L}{2} - \frac{\alpha S^2}{18N^2}\right)(X^{t+1} - X^t) + \frac{12\alpha S}{N}U^t + \frac{4\alpha S}{N}V^t + \frac{4\alpha L^2}{e}Y^t$$
$$+ (1+\omega)\alpha\sigma^2\left(\frac{65}{32K} + \frac{41}{SK}\right), \tag{40}$$

where we also plug in the choice of $\beta_1$ and $\beta_2$, use the fact that $N \ge S \ge 1$, and the stepsize condition $\alpha \le \min\{\frac{2}{9(1+\omega)}, \frac{1}{L}\}$ guarantees that $\frac{1}{2(1+\omega)\alpha} - \frac{L}{2} - \frac{S^3}{36N^3} \ge 0$.

Now define the Lyapunov function for all $t \in \mathbb{N}$,

$$\Psi^t := \mathbb{E}[f(x^t)] - f^\star + \frac{\alpha}{12}\mathbb{E}[\|\nabla f(x^{t-1})\|^2] + \left(\frac{1}{2(1+\omega)\alpha} - \frac{L}{2} - \frac{\alpha S^2}{18N^2}\right)X^t + \frac{24\alpha N}{S}U^t + \frac{4\alpha L^2}{e}V^t,$$

where $f^\star := \inf_x f(x)$. With the convention $x^{-1} = x^0$, the initial condition reduces to $\Psi^0 = R_1 + \frac{\alpha R_2}{12}$. Finally, substituting the definition of $\Psi^t$ into (40) and plugging the condition on $\alpha$ completes the proof. $\qquad\square$

# E  IMPLEMENTATION DETAILS AND ADDITIONAL NUMERICAL RESULTS

**Numerical results for ISCA.**  In Figure 4, we present the training loss for our algorithm (ISCA), SCAFFOLD, and FedAvg on MNIST and FashionMNIST under data homogeneous (IID) and heterogeneous (Non-IID) settings. The experimental settings in Figure 4 are the same as those in Figure 1. We observe from Figure 4 that, across two datasets, our algorithm demonstrates a faster convergence during the training process compared to the other two algorithms. Specifically, as we mentioned before, ISCA achieves similar test accuracy in both data homogeneous and heterogeneous settings. This uniformity in performance is mirrored by its consistent and rapid decrease in training loss, underscoring this robustness regardless of data heterogeneity. In contrast, FedAvg converges notably faster in the homogeneous setting. This observation ties directly into the test accuracy, where it performs significantly better in the homogeneous setting.

**Numerical results for ISCAM.**  In Figures 5 and 6, we plot the training loss for the algorithms under the same experimental settings as in Figures 2 and 3, respectively. ISCAM exhibits a notable advantage over the other two methods in terms of convergence speed. This rapid convergence, observed in both settings, aligns with ISCAM's strong performance in test accuracy, suggesting that the integration of the compressor does not impede the efficiency of ISCAM.

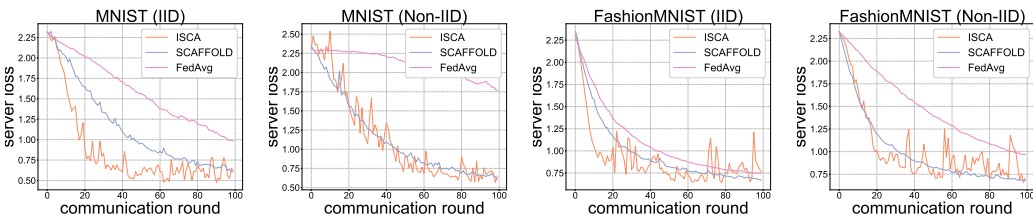

Figure 4: Training loss of our algorithm (ISCA), SCAFFOLD, and FedAvg under data homogeneous (IID) and heterogeneous (Non-IID) settings on MNIST (left half) and FashionMNIST (right half).

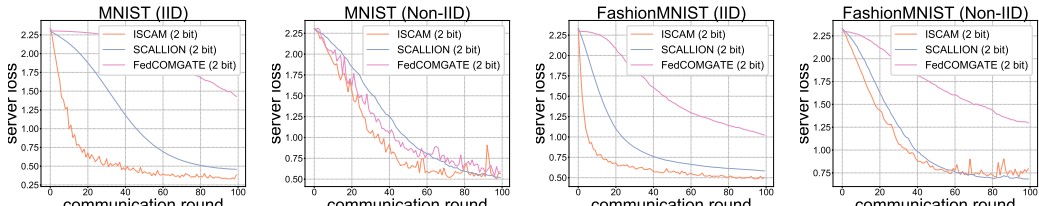

Figure 5: Training loss of ISCAM, SCALLION, and FedCOMGATE under data homogeneous (IID) and heterogeneous (Non-IID) settings, with 2-bit compression on MNIST (left half) and FashionM-NIST (right half).

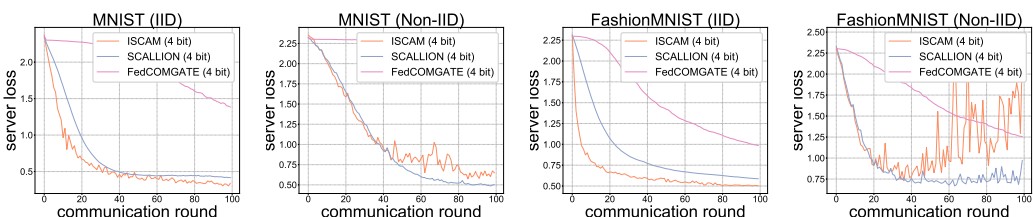

Figure 6: Training loss of ISCAM, SCALLION, and FedCOMGATE under data homogeneous (IID) and heterogeneous (Non-IID) settings, with 4-bit compression on MNIST (left half) and FashionM-NIST (right half).

