# OpenReview forum: "Improved Stochastic Controlled Averaging for Distributed and Federated Learning"
_ICLR.cc/2025/Conference — Submitted to ICLR 2025_

### Official Review · Reviewer_11fP · 2024-10-26

**Soundness:** 4
**Presentation:** 4
**Contribution:** 2
**Rating:** 5
**Confidence:** 3

**Summary:**

In this paper, the authors proposed a new algorithm to bridge the performance gap of existing methods in homogeneous and heterogeneous Federated Learning settings. Their algorithm is built upon SCAFFOLD and replaces 1) the ancient model at which the gradient is computed with an up-to-date model and 2) the fixed momentum with a momentum that is updated based on the local model. Then they provided a version of their algorithm with communication compression and derived convergence results for both of the two algorithms. Finally they conducted experiments to show that their proposed algorithms are unaffected by data heterogeneity.

**Strengths:**

- The theoretical results are thorough.
- The paper is well-written and easy to follow.

**Weaknesses:**

- The major concern I have is that the results are not strong enough to justify a new paper. In terms of theoretical results there is no improvement, and in the experiments the performance of the proposed algorithms underperform SCAFFOLD in the non-IID setting, even though the authors claimed that their algorithms are unaffected by data heterogeneity.
- The algorithms are not clearly presented. For example, if $u_{i}^{t+ 1, 0}$ is updated as Line 14 in Algorithm 1, it is unclear how we can obtain Lemma 8 in the appendix.
- The algorithms require computing the full gradient (Line 1 in Algorithm 1), which can be impractical in large-scale settings.

**Questions:**

- Please refer to Weaknesses
- The authors should provide numbers of experiment results in addition to the plots so that readers can have a better understanding of the improvements.
- The authors should provide evidence that the compression works, e.g., running time.
- Typos:
  - Line 222: $+\alpha_{\mathrm{in}}$ should be $-\alpha_{\mathrm{in}}$.
  - Line 225: "superscripts $i$ and $k$" should be "superscripts $t$ and $k$".

---

### Official Review · Reviewer_64xw · 2024-11-02

**Soundness:** 2
**Presentation:** 3
**Contribution:** 2
**Rating:** 5
**Confidence:** 2

**Summary:**

This work presents Improved Stochastic Controlled Averaging (ISCA) algorithm, proposed to address challenges in distributed and federated learning (D/FL). The authors claim that SCAFFOLD, a previous work in the literature, works well theoretically but shows inconsistent results in practical scenarios. ISCA aims to close this gap by updating local control variables more effectively during each iteration. The authors provide the theoretical convergence of ISCA and some numerical tests to assess its efficiency.
Additionally, the authors introduce ISCAM, which incorporates communication compression.

**Strengths:**

The main strengths are:

-The authors propose ISCA that addresses a limitation of SCAFFOLD by providing consistent performance across both homogeneous and heterogeneous data settings.

-The main novelty of ISCA is that it improves the local control variable update process, allowing it to better track changes in the local model at each iteration.

-The authors propose ISCAM, to handle communication compression. ISCAM extends ISCA  by incorporating unbiased communication compression.

-The authors show that ISCA maintains theoretical guarantees on convergence, comparable to SCAFFOLD, while demonstrating superior empirical performance. And also provide theoretical guarantees and empirical validation of ISCAM.

**Weaknesses:**

See my main concern in the next section,

**Questions:**

My main concern are:

-Besides the first subfigure in Figure 1, are there other references or examples that indicate SCAFFOLD performs poorly in IID settings?

-Have the authors conducted additional experiments in the IID setting to compare the performance of SCAFFOLD and ISCA?

-In lines 196-198, the author asserts that "...under a homogeneous setting ...f_1=...=f_N." This statement is not universally valid, as similarity in data does not imply that the objectives are equal. If the losses are indeed equal in a homogeneous setting, then why is there a need to consider finite sum losses? Wouldn't it suffice to focus on just one loss function in this case (e.g., f_1)?

- If one assumes that a homogeneous setting implies f_1=...=f_N, as the authors suggest, then in the case of using full gradients, the local control variables for all clients in SCAFFOLD would be identical. This would mean they do not influence the convergence towards the optimum. Could you explain why SCAFFOLD might perform poorly in such a setting? Have you conducted experiments under these conditions?

---

### Official Review · Reviewer_cVgs · 2024-11-02

**Soundness:** 2
**Presentation:** 2
**Contribution:** 1
**Rating:** 3
**Confidence:** 4

**Summary:**

The paper starts by an observation that SCAFFOLD, a celebrated FL algorithm, does not perform as well in the homogenous setting in FL. Motivated by this observation, the specific mechanism that is employed by SCAFFOLD to utilize control variates and thereby tame clients' drift is revisited. In doing so, a new algorithm is proposed and analyzed under standard assumptions. A standard extension to the compressed communication setting is further proposed.

**Strengths:**

While, the study of FL problems is crucial, the reviewer cannot point out to any specific strength pertaining to the results of the paper.

**Weaknesses:**

1. Given that almost all FL problems are heterogenous, it is unclear, if correct, how significant the observation that SCAFFOLD does not perform well in the homogenous setting is.  Can the authors point out an independent paper corroborating the observation "SCAFFOLD performs much better in the heterogeneous setting than in the homogeneous setting," as this seems contradictory to both the idea of control variates utilized by SCAFFOLD as well as the homogenous setting. In particular, it seems in the homogenous case, $c \approx c_i$ for all $i$, thus, the difference $x-y_{i}^{t,k+1}$ is simply $\alpha_{in}\sum_k \nabla F(y_i^{t,k})$. Now, since in the homogenous case, the clients have the same distribution, the term $\alpha_{in}\sum_k \nabla F(y_i^{t,k})$ should be nearly the same for all $i$, as long as the selection of clients and local minimi batches are i.i.d.

2. Upon comparing with Algorithm 1 in Karimireddy et al. (2020), it appears eq 1 and eq 2 in the paper are erroneous. In particular, $y_{i}^{t,k}+ \alpha_{in}$ and $+\frac{1}{K}(x^t-y_{i}^{t,k})$ in eq1 and eq 2 may need to be replaced with $y_{i}^{t,k+1}- \alpha_{in}$ and $+\frac{1}{K\alpha_{in}}(x^t-y_{i}^{t,k+1})$, respectively. Therefore, the reviewer is not convinced with the discussion in Section 3.2. Notably, the statement "Therefore, each active client uploads the newest local model and an
ancient local control variable to the server" seems inaccurate as, given the update of the control variate in SCAFFOLD, the total dirift is captured by the term $c_i^+-c_i$ which is communicated to the server.

3. While SCAFFOLD utilizes a process akin to SVRG and SPIDER, there are numerous new FL methods such as [a] and [b] that rely on more advanced approximate control variates methods such as STORM, which are shown to outperform SCAFFOLD in numerous scenarios. There are numerous important references missing from the paper. Two instances are listed below:

[a] Faster Non-Convex Federated Learning via Global and Local Momentum, UAI 2022

[b] Mime: Mimicking Centralized Stochastic Algorithms in Federated Learning, NeurIPS 2021

4. It appears the experimental setting is identical to Karimireddy et al. (2020) and Huang et al. (2024). However, there seem to be inconsistencies between Fig 1 (second and fourth plots) and Fig 1 (second and fourth plots) of Huang et al. (2024), notably regarding the performance of SCAFFOLD and FedAvg. Continuing the criticism regarding the experiments, the reviewer finds the extent limited, especially given that it is essential to corroborate the observation motivating this paper over an exhaustive list of models and datasets to convince the reader.

**Questions:**

Please see the questions raised in the Weakness section.

---

### Official Review · Reviewer_URZX · 2024-11-03

**Soundness:** 1
**Presentation:** 2
**Contribution:** 1
**Rating:** 3
**Confidence:** 5

**Summary:**

This paper addresses a shortcoming of SCAFFOLD, a well-known and popular algorithm for federated learning, namely: the practical performance of SCAFFOLD is heavily negatively impacted by increased heterogeneity. This is despite the theoretical guarantees for SCAFFOLD pointing towards heterogeneity having no impact on the algorithm's convergence rate. The authors identify two reasons for this inconsistency: (a) the use of old control variates to carry out the variance reduction, and (b) an inconsistency between the "momentum term" used and the stochastic gradient estimator used. In order to address this, they modify SCAFFOLD and develop a new algorithm, ISCA (Improved Stochastic Controlled Averaging for Federated Learning). They show ISCA's theoretical performance is, like SCAFFOLD, not impacted by data heterogeneity. Moreover, they also develop a variant of ISCA with compressed communication of gradients to further reduce the communication complexity. The paper finishes with experiments done on MNIST and Fashion MNIST with a fully connected neural network. The experiments show that ISCA performs well regardless of data heterogeneity.

**Strengths:**

The modifications attempted make sense, and it's very interesting to see how in practice SCAFFOLD is affected by data heterogeneity even though in theory it should not. I think in general trying to interrogate further the convergence of popular algorithms is useful, and rarely carried out.

**Weaknesses:**

There are several issues with this paper.

1. The primary motivation of this paper is to improve the practical performance of SCAFFOLD, since the theoretical guarantee given is no better than SCAFFOLD's. However, the experiments used are very toy: what works on MNIST is rarely what works on ImageNet, especially given that CNNs or Vision Transformers or other modern neural networks have several algorithmic components that interact in unexpected ways with the use of control variates, see e.g. [1]. If the goal of this paper is to improve the practical performance of SCAFFOLD, then the experimental evaluation should also be practical, with neural networks that people may actually use in practice.
2. The requirement on the inner stepsize from the theorem makes this stepsize so infinitesimally small we might as well be not doing any local steps. Also, the definition of $\alpha_{\mathrm{in}}$ seems to have some troublesome dimensions, i.e. by the requirement $144 \alpha_{in}^2 K^2 L^2 \leq \alpha = \alpha_{\mathrm{in}} \alpha_{\mathrm{out}} K$, we have that $144 \alpha_{\mathrm{in}} K L^2 \leq \alpha_{\mathrm{out}}$-- if we choose $\alpha_{\mathrm{out}} = 1$ (i.e. simple averaging, as in vanilla local SGD) we get that the inner stepsize ought to square with $1/L^2$, when the correct dimensional scaling would be with $1/L$. I tried finding where this comes from but
couldn't. The proofs are very difficult to read and somewhat disorganized. In line 1073 it's written "plugging the condition on $\alpha$ completes the proof but that's not true, the condition on $\alpha$ allows us to pick $\alpha_{\mathrm{in}}$ arbitrarily small, and then $\alpha$ itself arbitrarily small, which would make the $4R_1/(\alpha T)$ term explode. You need to actually specify the stepsize or assume both upper and lower bounds in terms of problem quantities.  Why is Lemma 4 applicable to Algorithm 1, when in Huang et al. (2024) it is derived for a different algorithm?
3. I don't get how Line 12 in Algorithm 1 is supposed to be computed on clients, when it involves the average function f. It seems to need the server to be computed, or there might be a typo.

Overall, I think this paper has many issues with both the practical application as well as the theoretical statements made, and therefore I can't recommend acceptance.

[1] Defazio, Aaron, and Léon Bottou. "On the ineffectiveness of variance reduced optimization for deep learning." Advances in Neural Information Processing Systems 32 (2019).

**Questions:**

Please address my concerns in the weaknesses section.

---

### Official Review · Reviewer_dzJw · 2024-11-04

**Soundness:** 2
**Presentation:** 2
**Contribution:** 2
**Rating:** 3
**Confidence:** 3

**Summary:**

The paper introduces the Improved Stochastic Controlled Averaging (ISCA) algorithm, designed to address the limitations of SCAFFOLD in federated learning, particularly in homogeneous data settings. ISCA aims to maintain the theoretical guarantees of SCAFFOLD while achieving improved empirical performance in both homogeneous and heterogeneous data environments. The paper also proposes ISCAM, an extension that incorporates communication compression and partial client participation.

**Strengths:**

The new algorithms (ISCA and ISCAM) show an attempt to improve upon SCAFFOLD, especially in homogeneous data settings, and the paper includes theoretical guarantees for the proposed methods, addressing convergence and model accuracy.

**Weaknesses:**

1. It is difficult to discern the clear differences between ISCA and SCAFFOLD in the methodology, particularly in terms of parameter updates and algorithmic steps.

2. There is no theoretical evidence to support ISCA as a better algorithm under homogeneous setting.

3. The improvements ISCA has made in homogeneous settings do not show practical significance in experiments and are, in fact, counterintuitive. The authors have not provided justification for the importance of homogeneous settings, especially when the prevailing trend in the field is to favor heterogeneous settings that more accurately align with real-world scenarios.

4. There is insufficient experimental evidence to validate the claimed benefits of ISCA, especially in non-IID settings and across diverse datasets. Specifically, the paper lacks comparisons with other FL algorithms like FedProx, FedNova, and on larger datasets such as FEMNIST, Shakespeare, and Reddit, making it difficult to judge ISCA’s relative performance.

**Questions:**

1. What is the difference between ISCA and SCAFFOLD, beyond local control variable adjustments? This distinction needs clearer explanation.

2. Could the authors clarify how ISCA enjoys performance benefits compared to SCAFFOLD while they both have the same convergence rate in Table 1?

3. Regarding Lemma 4 in Appendix B, how can the authors accommodate the existing results to ISCA? This needs further clarification.

4. I have some doubts on the testing performance. In Figure 1, MNIST (IID), SCAFFOLD has higher testing accuracy than ISCA initially, but then experienced weird performance decline. However, this trend never reappeared in other experiments.

5. Can additional baselines like FedProx and FedNova be included in Figure 1, particularly those addressing data heterogeneity?

---

### Meta-Review · Area_Chair_DqQ8 · 2024-12-16

**Metareview:**

This paper introduces two new algorithms, ISCA and ISCAM, designed to improve federated learning on heterogeneous datasets. The methods build upon SCAFFOLD and aim to address its convergence issues in such settings.

The reviewers appreciated the paper's identification of practical limitations in SCAFFOLD. They also acknowledged the promising trends observed in the numerical experiments for the proposed methods.

However, the theoretical results presented do not clearly improve on SCAFFOLD. The reviewers noted that the paper lacks sufficient insights and evidence to establish the effectiveness of the proposed changes convincingly.

The paper received five reviews, all consistent in their evaluation. Based on this consensus, I recommend rejection.

**Additional Comments On Reviewer Discussion:**

The authors did not submit a rebuttal, and the reviewers maintained their original evaluations of the paper without any changes.

---

### Decision · Program_Chairs · 2025-01-22

Reject